



# Consumption of atmospheric methane by the Qinghai–Tibetan

# Plateau alpine steppe ecosystem

**Hanbo Yun[1, 2, 3], Qingbai Wu[1*], Qianlai Zhuang[3*], Anping Chen[4]* Tong Yu[3], Zhou Lyu[3],**

**Yuzhong Yang[1], Huijun Jin[1], Guojun Liu[1], Yang Qu[3], Licheng Liu[3]**

**1. State Key Laboratory of Frozen Soil Engineering, Northwest Institute of Eco–**

**Environment and Resources, Chinese Academy of Sciences, Lanzhou, Gansu 730000,**

**China**

**2. Key Laboratory for Land Surface Process and Climate Change in Cold and Arid**

**Regions, Chinese Academy of Sciences, Lanzhou, 730000, China**

**3. Department of Earth, Atmospheric, and Planetary Sciences, Purdue University, West**

**Lafayette, Indiana 47907, USA**

**4. Department of Forestry and Natural Resources, Purdue University, West Lafayette,**

**Indiana 47907, USA**

**\*Authors for correspondence: qbwu@lzb.ac.cn [Q. W.],qzhuang@purdue.edu[Q.Z],**

**apchen1111@gmail.com [A.C.],**



## Abstract

Methane ($CH_4$) cycle on the Qinghai–Tibetan Plateau (QTP), the world's largest high–elevation permafrost region, is sensitive to climate change and subsequent cryoturbation dynamics. Yet its magnitudes, patterns, and environmental controls are still poorly understood. Here we report results from five continuous year–round $CH_4$ observation from a typical alpine steppe ecosystem in the QTP permafrost region. Results suggested the QTP permafrost region was a $CH_4$ sink of -$0.86 \pm 0.23$ g $CH_4$–C $m^{-2}$ $yr^{-1}$ over 2012 – 2016, a rate higher than that of many other permafrost areas such as Arctic tundra in northern Greenland, Alaska, and western Siberia. Soil temperature and soil water content were dominant factors controlling $CH_4$ fluxes and their correlations however changed with soil depths due to cryoturbation dynamics. This region was a net $CH_4$ sink in autumn, but a net source in spring, despite both seasons experienced similar top soil thawing and freeze dynamics. The opposite effect was likely caused by their season–specialized cryoturbation processes, which modified the vertical distribution of soil layers that are highly mixed like a multi–layer hamburger in autumn, but not in spring. Furthermore, the traditional definition of four seasons failed to capture the pattern of annual $CH_4$ cycle. We developed a new season division method based on soil temperature, bacteria activities, and permafrost active layer thickness, which significantly improved the modelling of annual $CH_4$ cycle. Collectively, our findings highlight the critical role of fine–scale climate and cryoturbation in driving permafrost $CH_4$ dynamics, which needs to be better monitored and modelled in Earth system models.



## 1. Introduction

Global atmospheric methane concentration [$CH_4$] resumed to rise since 2007 after
remaining stable between the 1990s and 2006 (Rigby et al., 2008; IPCC, 2013; Patra and Kort,
2016). Understanding mechanisms for this recent increase would require improved knowledge
on methane ($CH_4$) sources and sinks for regional and global $CH_4$ budget (Kirschke et al., 2013;
Zona et al., 2016). However, estimates on global $CH_4$ emissions and consumptions are still
highly uncertain (Spahni et al., 2011; Kirschke, 2013). In particular, the bottom–up approach,
which estimates $CH_4$ budget using ground observations and inventory, over–estimated the global
$CH_4$ budget by 6~20 times compared to the atmospherically constrained top–down approach
(Zhu et al., 2004; Lau et al., 2015). This discrepancy is partly due to limited monitoring data and
partly due to our poor understanding on important factors regulating the production and
consumption of $CH_4$ ( Whalen and Reeburgh, 1990; Dengel et al., 2013; Bohn et al., 2015).
The Qinghai–Tibetan Plateau (QTP), the world's largest high–elevation permafrost
region of $1.23 \times 10^6$ $km^{-2}$ (Wang et al., 2000), is currently experiencing a rapid change in climate
which affected cryoturbation processes, profoundly impacting methanotrophy and
methanogenesis and consequent net $CH_4$ fluxes (Mastepanov *et al.*, 2013; Lau et al., 2015).
However, due to the scarcity of high temporal–resolution and year–round environment and $CH_4$
monitoring, we still know little about the size, seasonal pattern, and underlying controls of
climate and permafrost cryoturbation and their effects on $CH_4$ exchanges in the QTP permafrost
region (Cao et al., 2008; Wei et al., 2015a; Song et al., 2015; ). This knowledge gap would also
hamper our capacity in predicting and understanding QTP permafrost $CH_4$ cycling under current
and projected future climates.





Here, we report results from a 5–year continuous *in situ* monitoring of $CH_4$ dynamics
with eddy covariance (EC) technique at the Beilu'he Research Station, a representative site for
QTP permafrost heartland covered by alpine steppe vegetation, from January 1st, 2012 to
December 31st, 2016. There are three advantages of our data acquisition system. First, the EC
system recorded data of $CH_4$ fluxes and climate, soil properties every half hour. As the QTP
permafrost is characterized with rapidly changing climate and soil cryoturbation dynamics even
over a short time period like within a day, different aerobic or anaerobic soil environments that
favor different types of $CH_4$ bacteria may also change frequently (Rivkina et al., 2004; Lau et al.,
2015). Thus, high–resolution *in situ* monitoring data would enable us to quantify $CH_4$ exchange
patterns from diel to annual time–scales and investigate their major environmental drivers.
Second, our field investigation spanned five full calendar years including both plant growing and
non–growing seasons. Observations of the plant non–growing season that account for two-thirds
of a year were very rare in current literature (Song et al., 2015). Third, the EC system we used
overcomes some technical problems caused by the previously often used static chambers,
including limited representation of local site heterogeneity and additional heating of the soil
surface (Chang et al., 2014; Wei et al., 2015b).
**2. Methods**
**2.1 Site Description**
The research site, Beilu'he permafrost research station (34° 09' 006" N, 92° 02' 080" E),
is located in the alpine steppe continuous permafrost area of the northern QTP, about 320
kilometers southwest of Golmud (Figure 1). At an elevation of 4765 meters, the air is thin with
only 0.6 standard atmospheric pressure. According to *in situ* observations, the site receives solar
radiation about 6720 MJ m$^{-2}$. The non–growing season is long and cold with 225 days per year



and annual air temperature is -18 ℃ on average from 2012 to 2016.  Its growing season is short
and cool with 140 days per year from 2012 to 2016 and mean annual air temperature is 4.6 ℃.
According to the site drilling exploration, the permafrost depth can extend to $50 - 70$ m
belowground, and the thickness of the active layer (ALT) is about $2.2 - 4.8$ m. The soil is
composed of Quaternary fine sand or silt (Table 1), overlying on Triassic mudstone or weathered
marl. Dominant plant species includes *Carex moorcroftii Falc. ex Boott*, *Kobresia tibetica*
*Maxim*, *Androsace tanggulashanensis*, *Rhodiola tibetica*. Vegetation coverage is approximately
33.5% and average plant height is 15 cm.
**2.2 Eddy Covariance observations**

We have continuously monitored $CH_4$, carbon dioxide ($CO_2$), water ($H_2O$) and heat flux

using a standard eddy covariance system tower 3m above the ground. $CH_4$ flux was measured
with an open-path $CH_4$ analyzer system (Figure 1: d; LI–7700, LI–COR Inc., Lincoln, NE, USA).
The precision is 5 ppb RMS noise at 10 Hz and 2000 ppb. The instrument was placed on site on
August 8[th], 2011, and then connected to a three–dimensional sonic anemometer (heat and water
flux; CSAT3, Campbell Scientific, and Logan, UT, USA; precision is 0.1 ℃; accuracy is within 1%
of reading for half–hour) and an open–path infrared gas analyzer ($CO_2$ flux; LI–7500A, LI–COR
Inc., Lincoln, NE, USA; the precision is $0.01 \mu mol \ m^{-2} \ s^{-1}$ and the accuracy is within 1% of
reading for half–hour, zero drift per ℃ is $\pm$ 0.1 ppm typical) on January 1[st], 2012 when the
system worked steadily. Monitoring data were recorded and stored 10 Hz using a data logger
(LI–7550, LI–COR Inc., Lincoln, NE, USA).

To reduce the LI–7500A surface heating / cooling influence on $CO_2$ and $H_2O$ molar

densities in tough environments. Each year, "summer style" was used in Li–7500A, in which



surface temperature setting is 5℃ during May 1st to September 30th; "winter style" was used from
October 1st to the next year April 30th in Li–7500A, in which surface temperature setting is -5℃.
Calibrations of $CO_2$, water vapor, and dew point generator measurements for LI–7500A
analyzers were performed regularly by the China Land–Atmosphere Coordinated Observation
System (CLAROS). Up–and–down mirrors of LI–COR 7700 were cleaned regularly every 30
days to make sure the signal strength was stronger than 80. All of these instruments were
powered by solar–panel and battery.

**2.3 Micrometeorological and Soil Measurements**

A wide range of meteorological variables was measured by a standard automatic
meteorological tower 3 m above the ground and northern 5 m apart from the eddy covariance
tower. Rn and albedo were measured with a four–component radiometer (Rn; CNR–1, Kipp and
Zonen, the Netherlands). Air temperature ($T_{air}$), air relative humidity, and atmospheric pressure
were measured with a temperature and humidity sensor (HMP45C, Vaisala Inc., Helsinki,
Finland) in the meteorological tower. A rain gauge (TE525MM, Texas Electronics Inc., Dallas,
TX, USA) was used to measure precipitation process. Wind speed and wind direction were
observed using a propeller anemometer placed on the top of the meteorological tower.
Two self–calibrating soil heat flux (SHF) sensors (HFP01) were placed 5 cm and 15 cm
below the ground. A group pF–Meter sensor (GEO–Precision, Germany) was embedded in the
soil under meteorological tower to measure soil temperature ($T_{soil}$) at 0 cm, 5 cm, 10 cm, 15 cm,
20 cm, 30 cm, 40 cm, 50 cm, 70 cm, 80 cm, 100 cm, 150 cm, 160 cm, 200 cm depth and soil
relative water content (SWC) at 10 cm, 20 cm, 40 cm, 80 cm, and 160cm depth. Both the air
temperature, humidity sensors, and pF meter sensors were calibrated in the State Key Laboratory



of Frozen Soil Engineering of the Chinese Academy of Sciences to ensure the measurement
accuracy is within ± 0.05 °C and ± 5%, respectively.

All of above environment parameters were synchronously monitored with eddy covariance

and the data were recorded every 30 minutes by CR3000 (Data logger, Campbell Data Taker Ltd,
Salt Lake City, UT, USA).

In August 2010, one 1 m × 1 m × 2 m pit was dug for soil sample collection and installation

of soil environmental sensors, it is 10 m from the eddy covariance tower. Five profile samples
were taken from the pit at depths 0 − 20 cm, 20 − 50 cm, 50 − 120 cm, 120 − 160 cm, and
160 − 200 cm. Every depth was repeated for five times, after fully mixed, then stored in soil
sample aluminum boxes and carefully sealed to prevent gas exchanges with air. The clod method
was used to investigate the field wet bulk density (weight of soil per unit volume; Cate and
nelson, 1971). The soil moisture content was calculated gravimetrically by the ratio of the mass
of water present to the oven–dry (60 ℃, 24 h) weight of the soil sample. Soil organic carbon
(SOC) content of the air–dried soil samples was analyzed using the wet combustion method,
Walkley–Black modified acid dichromate digestion, $FeSO_4$ titration, and an automatic titrator.
Total Nitrogen (TN) and pH were measured using standard soil test procedures from the Chinese
Ecosystem Research Network.

Following Muller's original definition, ALT is the maximum thaw depth in the late

autumn using a linear interpolation of Tsoil profiles between two neighboring points above and
below the 0 °C isotherm (Muller, 1947). We used records of the soil thawing thickness measured
with a self–made geological probe to verify the ALT data semi–monthly. More information
about the measurement procedure was previously described (Wu and Zhang, 2010a).



**2.4 Microbial Activity**

We sampled 100 gram soils by soil sample drill device (Ø=0.03 m) from $0 - 25$ cm depth every 5 days. The sampled soil was fully mixed and divided into two equal parts, storing in sterilized aluminum box placed in liquid nitrogen before sending to the lab for microbe RNA extraction. We then used a real-time PCR method to genetically test r methanotrophic / archaeal methanogens, and the procedure was repeated three times for each sample. By setting the maximum methanotrophic / archaeal methanogens gene expression cyclic number as 1, we calculated the variety coefficient of methanotrophic and archaeal methanogens gene expressions ($\Delta$I and $\Delta$II, respectively; %) with equation (1) :

$$\Delta_i = {x_I}/{X_{Max}} \qquad \ldots \qquad (1)$$

$\Delta_i$ is for the $i^{th}$ methanotrophic/archaeal methanogens gene expression; $x_i$ is the methanotrophic / archaeal methanogen gene expression cyclic number of the $i^{th}$ time; $X_{Max}$ is the maximum methanotrophic / archaeal methanogen gene expression cyclic number of the soil group from 2012 to 2016.

**2.5 EC Data Processing and Data Filtering**

Data collected from January $1^{st}$, 2012 to December $31^{st}$, 2016 were used in this study. Before processing, we removed data that were recorded at the time of precipitation events or with LI–7700 signal strength under 85. We first processed the raw data in Eddypro 6.2.0 (LI–COR, Lincoln, NE, USA). We adopted standardized procedures recommended in Lee et al. (2006) to process half–hourly flux raw measurements to ensure their quality:





1) data were processed through statistical analysis in Eddypro 6.2.0 including: spike
removal (accepted spikes < 5% and replaced spikes with linear interpolation), amplitude
resolution (range of variation: 7.0 σ, number of bins: 100, accepted empty bins: 70%), drop–outs
(percentile defining extreme bins: 10, accepted central drop–outs: 10%, accepted extreme drop–
outs: 6%), absolute limits (-30 m s$^{-1}$ < U < 30 m s$^{-1}$, -5 m s$^{-1}$ < W < 5 m s$^{-1}$, -40 ℃ < Ts < 40 ℃,
200 µmol mol$^{-1}$ < $CO_2$ < 500 µmol mol$^{-1}$, 0 µmol mol$^{-1}$ < $H_2O$ < 40 µmol mol$^{-1}$, 0.17 µmol < $CH_4$
< 1000 µmol), Skewness and kurtosis (-2.0 < Skewness lower limit < -1.0, 1.0 < Skewness up
limit < 2.0; 1.0 < Kurtosis lower limit < 2.0, 5.0 < Kurtosis upper limit < 8.0), discontinuities
(hard–flag threshold: U = 4.0, W = 2.0, $T_S$ = 4.0, $CO_2$ = 40, $CH_4$ = 40, and $H_2O$ = 3.26; soft–flag
threshold: U = 2.7, W = 1.3, $T_S$ = 2.7, $CO_2$ = 27, $CH_4$ = 30, and $H_2O$ = 2.2), angle of attack
(minimum angle of attack = -30, maximum angle attack = 30, accepted amount of outliers =
10%), steadiness of horizontal wind (accepted wind relative instationarity = 0.5) (Vickers and
Mahrt, 1997; Mauder et al., 2013).
2) The data were then corrected using atmosphere physical calculation expressed by: axis
rotations of tilt correction (double rotation), time lags compensation (covariance maximization),
and compensating density fluctuations of Webb–Pearman–Leuning (WPL) terms. When $CO_2$ and
$H_2O$ molar densities are measured with the LI–COR 7500 / LI–COR 7500A in cold
environments (low temperatures below -10 ℃), a correction should be applied to account for the
additional instrument–related sensible heat flux, due to instrument surface heating / cooling.
Thus, we implemented the correction according of Burba et al. (2008), which involves
calculating a corrected sensible heat flux ($H'$) by adding estimated sensible heat fluxes from key
instrument surface elements, including the bottom window ($H_{bot}$), top window ($H_{top}$), and spar
($H_{spar}$)—to the ambient sensible heat flux ($H$):



$H' = H + H_{bot} + H_{top} + 0.15 \times H_{spar}$      …      (2)
3) Quality assurance (QA) / quality control (AC) were ensured through spectral analysis
and corrections analysis in Eddypro 6.2.0: spectra and co–spectra calculation used power–of–two
samples to speed up the Fast Fourier Transform (FFT) algorithm. Spectra and co–spectra QA /
AC by filter (co)spectra were made according to Vickers and Mahrt (1997) test results, and
micrometeorological quality test results (Mauder and Foken, 2004). Low–frequency range
spectral correction was done considering high–pass filtering effects. High–frequency range
spectral correction was done considering low–pass filtering effects (Moncrieff et al., 2004).
4) We chose values "0","1","2" to flag the processed flux data into three quality classes
in Eddypro 6.2.0. The combined flag attains the value "0" for best quality fluxes, "1" for fluxes
suitable for general analysis such as annual budgets and "2" for fluxes that should be discarded
from the results dataset.  For our dataset, approximately 67% of the data fell into Class 0, 12% in
Class 1, and 21% in Class 2.
5) Our analysis indicated that, under average meteorological conditions, 80% of the flux
came from area within 175 m of the eddy covariance tower.
In addition, we also adopted the method in Burba et al. (2008) to adjust the half–hour flux
data to avoid apparent measuring errors. In doing this, we rejected half–hour flux data that fall
into one of the following situations: (1) incomplete half–hour measurements, (2) measurements
under rain impacts, (3) nighttime measurements under stable atmospheric conditions (U*,
friction velocity, < 0.1 m s$^{-1}$), and (4) abnormal values detected by a three–dimensional
ultrasonic anemometer. This screening resulted in the rejection of about 20.7% of the flux data.





After the above data quality control, there was a 28.7% data gap for CH$_4$ fluxes over the
entire exanimation period. These data gaps were then filled according to the method described in
literature (Falge et al., 2001; Papale et al., 2003). We used a linear interpolation to fill the gaps if
they were less than 2 hours, a method described in Falge *et al.* (2001) to fill gaps greater than 2
hours but less than 1 days, and an artificial neural network approach as described in Papale et al.
(2003) and Dengel et al. (2013) to fill gaps greater than 1 day.
The quality of the dataset was evaluated using the equation of energy closure:
$$EBR = \sum (H + \lambda E) / \sum (R_n - G - S) \quad \ldots \qquad\qquad 3$$

where the *EBR* is surface energy balance ratio; *H* is heat flux; $\lambda E$ is latent heat; *Rn* is net
radiation; *G* is SHF; and *S* is heat storage of vegetation canopy. As vegetation coverage at this
research site is sparse, *S* is ignored. From 2012 to 2016, the *EBR* average was larger than 67.5%.
We analyzed two different major sources of CH$_4$ flux gap–filling uncertainty: the first
kind of uncertainty came from U* threshold estimate. Following Burba et al. (2008), we
excluded the probably false low CH$_4$ flux at low U*, but how to decide the U* threshold still
remained highly uncertain. For instance, when choosing a lower U* threshold, the associated
lower flux would contribute to the gap filling and the annual gross (Loescher, et al., 2006). The
variance from 5% to 95% of bootstrap provided an average of the uncertainties caused by the U*
filter out. The second uncertainty source was due to insufficient power supply. In this research,
all instrument power was supplied by solar. Long-time rainy, cloudy, and snow weather, would
cause the instrument to stop working by insufficient power supply. When we used the method to
fill the gap mentioned above, it would cause the CH$_4$ deviated from the true value.   To our



knowledge, the $CH_4$ flux data were with large uncertainty under rainy conditions.
**2.7 Statistical Analyses**
We performed correlation, principal component analyses (PCA) and linear regression
analyses in IBM SPSS (IBM SPSS Statistics 24; IBM, Armonk NY, USA). Specifically, we used
bivariate correlation to examine pairwise relationships between environmental factors and $CH_4$
fluxes, PCA and linear regressions to explore the sensitivity of $CH_4$ fluxes to simultaneous
environmental fluctuations in wind speed, Tair, air relative humidity, Rn, vapor pressure deficit
(VPD), albedo, SHF, SWC, and Tsoil. Before performing PCA and linear regressions, the entire
dataset was examined for outliers (Cook's Distance, < 0.002), homogeneity of variance (Levene
test, $p < 0.05$), normality (Kolmogorov–Smirnov test, smooth line for histogram of Studentized
residuals), collinearity (variance inflation factor, $0 < VIF < 10$), potential interactions ($t$–test, $p <$
$0.05$), and independence of observations ($t$–test, $p < 0.05$).
We performed structural equation modeling (SEM) to evaluate the effects of
environmental variables on $CH_4$ fluxes for different seasons. SEM is a widely-used multivariate
statistical tool that incorporates factor analysis, path analysis, and maximum likelihood analysis.
This method uses *priori* knowledge on the relationships between focus variables to verify the
validity of hypotheses. Here we performed SEM analyses with AMOS 21.0 (Amos Development
Corporation, Chicago, IL, USA).  All data are presented as mean values with standard deviations.
**3. Results**
**3.1 Meteorological Conditions**
Environmental factors were observed according to meteorological records from the
Beilu'he Permafrost Weather Station from 2012 to 2016. Mean annual Tair was -4.5 ℃ (Figure



2), with minimum and maximum mean diel temperatures of -21.6 ℃ (12[th] January, 2012) and
13.8 ℃ (28[th] July, 2015), respectively. Average net radiation (Rn) was 82.8 Wm$^{-2}$, while the
maximum was in August (136.2 Wm$^{-2}$; Figure 3). The average VPD was about 0.3, while the
maximum was 0.98 and the minimum was 0.02 (Figure 4). Mean annual precipitation was 335.4
mm (Figure 5), which was primarily based on rain and snowfall (only occupied 7%). It is very
different from the high-latitude permafrost area. From 2012 to 2016, the maximum precipitation
was 2013 (488.3 mm), and the minimum was the 2015 (310.0 mm). The main contribution of
precipitation was in summer, about 92%. During winter, the precipitation was rare and the mean
value was about 6.7 mm, and even decreased from 14.2 mm in 2012 to 2.1 mm in 2016. Spring
was another important rainfall period besides summer, in which mean precipitation was about
37.5 mm, 8～17% of the total.

Figure 6 and Figure 7 showed the SWC and Tsoil varieties of soil layers from 2012 to

2016, respectively. Mean SWC of depths 10 cm, 20 cm, 40 cm, 80 cm, and 160 cm were 14%,
9%, 8%, 14% and 19%, respectively. Tsoil of depths 0 cm, 5 cm, 10 cm, 20 cm, 30 cm, 40 cm,
50 cm 70 cm and 80 cm were along with the Tair changes, but at depths 100 cm, 150 cm, 160 cm
and 200 cm were not.  The Tsoil of depth 200 cm had a remarkable difference from Tsoil of
other layers. The reason could be that some peats exist in this layer and during winter the peat
layer were not completely frozen.  Figure 8 showed that SHF half-hour and diel scale varieties of
5 cm and 15 cm depth. Annual mean value of SHF at 5 cm and 15 cm depth is 7.6 Wm$^{-2}$ and 6.8
Wm$^{-2}$, respectively.

The Beilu'he has a windy environment (Figure 9). Its annual average speed was 4.4 m s$^{-1}$

from 2012 to 2016, while its maximum and minimum wind speeds were 14.6 m s$^{-1}$ on 14[th]



February, 2016 and 1.3 m s$^{-1}$ on 1$^{st}$ November, 2013, respectively. Its winter, spring, and autumn
average wind speed were 5.4 m s$^{-1}$, 4.3 m s$^{-1}$, and 3.7 m s$^{-1}$, respectively, while the principal
direction of the strongest winds were from the southwest. Late autumn, winter, and early spring
drought increased risks of dust blowing days of 122 within a year on average. Its summer
average wind speed was about 3.30 m s$^{-1}$, with the southeast wind dominated.
Figure 10 illustrates the processes of soil freezing and thawing observed from January
2012 to December 2016. The duration of the active layer in the thawing state at 40 cm depth
ranged from 174 to 188 days with an average variation of up to 14 days. The average ALT is 4.4
m from 2012 to 2016.

**3.2 New classification system of the four seasons**

Based on microbial activities (Figure 11), ALT variety coefficients (ALT variety
coefficient = (ALT$_{i+1}$ - ALT$_i$) / ALT$_{Max}$, where ALT$_{Max}$ is the maximum of ALT per year), and
Tsoil, we re–defined the four seasons of spring_, summer_, autumn_, and winter_. Below we
describe the start date of each season. The end date of a season is the day immediately before the
start of the next season. Spring_ starts at the first day of two consecutive observation periods
fulfilling both (1) ($\Delta II + \Delta I$) / 2 $\geqslant$ 15%, and (2) the ALT variety coefficient $\geqslant$ 0.05. Summer_
starts on the first day of two consecutive observation periods when (1) ($\Delta II + \Delta I$) / 2 $\geqslant$ 45%, (2)
ALT variety coefficient $\geqslant$ 0.35, and (3) five successive days with Tsoil at 40 cm soil depth $\geqslant$
0 ℃. Autumn_ starts on the first day of two consecutive observation periods when (1) ($\Delta II + \Delta I$)
/ 2 $\geqslant$ 55%, (2) the ALT variety coefficient $\geqslant$ 0.60, and (3) five successive days the Tsoil of 10
cm $<$ 5 ℃. Winter_ starts on the first day of two consecutive observation periods that (1) ($\Delta II +$
$\Delta I$) / 2 $<$ 15% and the ALT variety coefficient $<$ 0.05.





To test the robust of our new division method of seasons in our methane cycle analysis,
we compared empirical $CH_4$ flux estimates using different season definitions (Table 2). In
addition to our new method that was based on top soil microbe activity, Tsoil of 0 – 40 cm, and
permafrost active layer variability (hereafter refer to as SMT), we also used three conventional
methods–one on vegetation cover and temperature change (VCT), one on Julian months (JMC),
and the other one on vegetation phenology change (VPC). Specifically, the VCT method splits a
year into plant growing season and non–growing season; the JMC method assumes May to
October as plant growing season, and November to the following April as non–growing season;
and the VPC method defines plant growing season as the period between when all dominant
grass species (*Carex Moorcroft Falc. ex Boott*, *Kobresia tibetica Maxim*, *Androsace*
*tanggulashanensis*, *Rhodiola tibetica*) germinate and when they all senesce. For each of the four
methods, we established empirical maximum likelihood models between all environmental
factors and diel $CH_4$ fluxes over each season and then compared modeled $CH_4$ fluxes and field
observations under those methods of different seasonal definitions (Figure 12). We found that the
agreement between modeled and observed $CH_4$ fluxes using the new SMT method reached $R^2 =$
0.28, almost twice as that of the VPC ($R^2 = 0.17$) and VCT ($R^2 = 0.14$) methods, and more than
three times that of the JMC method ($R^2 = 0.08$) (Figure 12). Hence, the comparison suggested
that our new method could better model $CH_4$ fluxes over a year. The use of traditional plant
growing season versus non–growing season definitions may also underestimate or overestimate
$CH_4$ sinks or sources, especially when many studies assume $CH_4$ close to zero during the plant
non–growing season. Furthermore, the new SMT method well captures the impact of spring_m
and autumn_m permafrost thawing / freezing cycles on $CH_4$ fluxes, and the different preferable





environments for methanogens and methanotrophic bacteria during the summer_m season, while
conventional methods do not.
**3.3 Annual, Seasonal and Diel Variabilities of Methane Fluxes**
Our results indicated that the Beilu'he site was a $CH_4$ sink with an annual mean strength
of -0.86 ± 0.23 g $CH_4$–C $m^{-2}$ (95% confidence interval; negative values mean $CH_4$ sinks, positive
values mean $CH_4$ sources), although the strength of the $CH_4$ sink varies across different years
from -0.57 ± 0.27g $CH_4$–C $m^{-2}$ $yr^{-1}$ in 2015 to -1.49 ± 0.38g $CH_4$–C $m^{-2}$ $yr^{-1}$ in 2014 (Figure 5).
The amount of gene expression by methanogens and methanotrophs at 0 − 25 cm soils in March
and November, for instance, were about 16.8% and 35.6%, respectively, suggesting strong
microbial activities even during the cold and dry plant non−growing season (Figure11).
We also observed clearly $CH_4$ seasonal variations (Figure 13), in both the amount of $CH_4$
exchanges and their diel cycles (Figure 14). In spring_, the Beilu'he site was a $CH_4$ source of
0.90 ± 0.37 g $CH_4$–C $m^{-2}$ $yr^{-1}$ (Figure 13: a), accounting for 53% of annual $CH_4$ emissions, or
1.81 ± 0.22 mg $CH_4$–C $m^{-2}$ $d^{-1}$. For a typical spring_ (Figure 14: a2, b2, c2, d2, and e2), diel $CH_4$
emission usually started at around 10:00 am ∼ 10:30 am, when the thin ice layer on the soil
surface started to thaw. It then reached the peak at 12:30 pm ∼ 13:30 pm. The emission peak
started to weaken at around 15:30 pm ∼ 16:00 pm, and reached around zero or even turned into
a small sink after 20:00 pm.
In summer_, the Beilu'he site was a $CH_4$ sink of -0.99 ± 0.18 g $CH_4$–C $m^{-2}$ $yr^{-1}$ (Figure
13: b), or -13.28 ± 0.38 mg $CH_4$–C $m^{-2}$ $d^{-1}$. The diel cycle of $CH_4$ fluxes in summer_ was
characterized with two absorption peaks and one small emission peak (Figure 14: a3, b3, c3, d3,
and e3). With Tair increasing after sunrise, soils started to absorb atmospheric $CH_4$ and this soil



uptake process reached its first peak at around 9:30 am ～ 10:30 am. After then the continuously
increasing Tair turned to suppress $CH_4$ uptake and promote $CH_4$ emissions, likely due to
different temperature sensitivities of methanotrophic and methanogenic bacteria. At around
15:30pm ～ 16:00 pm when Tair reached the maximum (Figure 2: b), $CH_4$ emission also reached
its peak. The following temperature decrease in the late afternoon again reversed the $CH_4$ uptake
/ emission process, and by sunset, we observed another $CH_4$ sink peak. The rate of $CH_4$ sink then
decreased again through the night with further decreasing temperature.

Autumn_ was another season with net $CH_4$ sink and even had the largest amount of $CH_4$

sink in 2013 (Figure 13: c). The $CH_4$ sink in autumn_ varied between -0.69 ± 0.19 g $CH_4$–C $m^{-2}$
(2015) and -1.59 ± 0.33 g $CH_4$–C $m^{-2}$ (2013), with an average diel rate of -1.19 ± 0.48 g $CH_4$–C
$m^{-2}$ $yr^{-1}$ or -13.31 ± 0.28 mg $CH_4$–C $m^{-2}$ $d^{-1}$. The diel dynamics of autumn_ $CH_4$ fluxes was like a
letter "V", with a single sink peak during 13:30 pm ～ 15:30 pm (Figure 14: a4, b4, c4, d4, and
e4).

In winter_, the net $CH_4$ flux at the Beilu'he site was an atmospheric source, with an

average annual rate of 0.41 ± 0.16 g $CH_4$–C $m^{-2}$ $yr^{-1}$ or 4.35 ± 0.33 mg $CH_4$–C $m^{-2}$ $d^{-1}$ (Figure 13:
d). It also should be noted that since the investigation started from January $1^{st}$, 2012 and ended
on December $31^{st}$, 2016, the 2011 ～ 2012 and 2016 ～ 2017 winters_ were only about half of
the regular length. The diel $CH_4$ cycle of an average winter_ day was characterized by one single
emission peak around 10:30am ～ 17:30 pm (Figure 14: a1, b1, c1, d1, e1 and f1).
**3.4 Response to changes in Methane to Environmental Factors**

Diel fluxes of $CH_4$ were highly correlated with many biotic and abiotic environmental

factors, either positively or negatively (Table 3). Positive factors include metagenomics of both





methanotrophic (r = 0.52, $p < 0.01$) and methanogens (r = 0.49, $p < 0.01$) at $0 - 25$ cm soils,
ALT (r = 0.43, $p < 0.01$), and wind speed (r = 0.15, $p < 0.01$). Important negative factors include
VPD (r = -0.26, $p < 0.01$), SWC at all depths (varied r values between -0.17 and -0.26, $p < 0.01$),
Tair (r = -0.11, $p < 0.01$), and air pressure (r = -0.15, $p < 0.01$). The correlation signal between
$CH_4$ fluxes and Tsoil changed with soil depths (varied r values between -0.09 and 0.24, $p < 0.01$).
Furthermore, path analysis results showed that Tsoil at 5cm and 10cm were the most important
factors, which together contributed about 25% of the relative importance coefficient. Following
were SWC at 80 cm (14%) and 20 cm (12%), and Tsoil at 20 cm (8%).

Further analyses suggested that dominant control factors of $CH_4$ fluxes also changed

among different seasons. In spring_, Rn was the most important factor with a relative importance
coefficient near 60%, followed by SHF at 5 cm (9%) and SWC at 20 cm (6%). Table 4 shows the
results of PCA. In spring_, PC1 explained 63% of the $CH_4$ variations, which was positively
correlated with Tair, VPD, Rn, SHF of 15 cm, ALT, ΔI, SWC of $10 - 40$ cm, Tsoil of 0 cm,
Tsoil of $5 - 20$ cm, Tsoil of $30 - 50$ cm, and negatively correlated with wind speed. The PC2
explained about 23% of $CH_4$ fluxes variations. PC2 was positively correlated with wind speed,
Tair, Rn, SHF of 15cm, but negatively correlated with VPD, ALT, ΔI, SWC $10 - 40$ cm, Tsoil of
0 cm, Tsoil of $5 - 20$ cm, and Tsoil of $30 - 50$ cm. The first four principal components explained
about 86% of the $CH_4$ variations.

In summer_, the relative importance coefficient of Tsoil at 100 cm and 200 cm was about

30.2% and 26.5%, respectively, followed by Tsoil at 70 cm (12.3%) and Tsoil at $0 - 20$ cm
(11.4%). The first four principal components explained about 88% of the $CH_4$ variations (Table
4). PC1 explained 70% of the $CH_4$ variations. PC1 was positively correlated with wind speed,
Tair, VPD, SHF of 15 cm, ALT, ΔI, SWC of $50 - 160$ cm, precipitation, Tsoil of 0 cm, Tsoil of 5



− 40 cm, Tsoil of 50 − 80 cm, and Tsoil of 100 − 200 cm, but negatively correlated with Rn, and
SWC of 10 − 40 cm. PC2 was positively correlated with wind speed, Tair, VPD, Rn, SHF of
15cm, SWC of 10 − 40 cm, Tsoil of 0 cm, but negatively correlated with ALT, ΔI, SWC of 50 −
160 cm, precipitation, Tsoil of 5 − 40 cm, Tsoil of 50 − 80 cm, and Tsoil of 100 − 200 cm.

In autumn_, Rn and Tsoil at 5 − 20 cm had the highest relative importance coefficients

(18.3%), for Rn and Tsoil is 11.5% and 16.7%, respectively. The first four principal components
explained about 86% of the CH$_4$ variations (Table 4). PC1 explained 69% of the CH$_4$ variations.
PC1 was positively correlated with Tair, VPD, Rn, SHF of 15 cm, ALT, ΔI, SWC of 10 − 40 cm,
SWC of 50 − 160 cm, Tsoil of 0 cm, Tsoil of 5 − 40 cm, Tsoil of 50 − 80 cm, and Tsoil of 100 −
200 cm, but negatively correlated with wind speed. PC2 was positively correlated with wind
speed, Tair, Rn, SHF of 15 cm, ALT, ΔI, Tsoil of 0 cm, and Tsoil of 5 − 40 cm, but negatively
correlated with VPD, SWC of 10 − 40 cm, SWC of 50 − 60 cm, Tsoil of 50 − 80 cm, and Tsoil
of 100 − 200 cm.

During winter_, Rn was again the most important factor (34% relative importance

coefficient), followed by Tsoil at 0 − 40 cm (27% in total), and SHF of 15 cm (17% in total). The
first four principal components explained about 96% of the CH$_4$ variations (Table 4). PC1
explained 75% of the CH$_4$ variations. PC1 was positively correlated with wind speed, Tair, VPD,
Rn, SHF of 15 cm, ΔI, Tsoil of 0 cm, and Tsoil of 5 − 20 cm. PC2 explained 21% of the CH$_4$
variations. PC2 was positively correlated with wind speed, Tair, Rn, SHF of 15 cm, and ΔI, but
negatively correlated with VPD, Tsoil of 0 cm, and Tsoil of 5 − 20 cm.
**4. Discussion**



**4.1 New Classification System of the Four Seasons**

Here, different from the majority of earlier studies (Treat et al., 2014; Wang et al., 2014;

Wei et al., 2015a; Song et al., 2015), we adopted a new classification system of the four seasons

based on soil bacteria activities, Tsoil of $0 - 40$ cm and ALT, rather than the conventional

methods based on Tair and vegetation dynamics (Chen et al., 2011; McGuire et al., 2012).

Previous studies indicated that changes in $CH_4$ fluxes are regulated by soil microbes, and

activities of soil microbes are not limited to the warm season (Zhuang et al., 2004; Lau et al.,

2015; Yang et al., 2016). For instance, in March and November, we found the amount of gene

expression by methanogens and methanotrophs at $0 - 25$ cm soils were about 16.8% and 35.6%

(Figure 11), respectively, suggesting there are still strong microbial activities during the cold and

dry season. Therefore, our new method of defining the four seasons from the top soil's biotic and

abiotic features shall better capture the pattern of $CH_4$ dynamics throughout a year.

**4.2 Annual, Season mean and Diel Variability**

From 2012 to 2015, the annual mean value was $-0.86 \pm 0.23$ g $CH_4 - C$ m$^{-2}$ of the alpine

steep ecosystem in Beilu'he. This sink strength is larger than that of previous reports from other

sites of the QTP (Cao et al., 2008; Wei et al., 2012; Li et al., 2012; Song et al., 2015; Chang and

Shi, 2015) and many high − latitude Arctic tundra ecosystems like northeast Greenland

(Jørgensen et al., 2015), western Siberia (Liebner et al., 2011), and Alaska (Whalen et al., 1992;

Zhuang et al., 2004; Whalen, 2005). Different hydrothermal conditions, which greatly influence

$CH_4$ cycling in permafrost regions (Spahni et al., 2011; Kirschke et al., 2013), may partly explain

the site difference in $CH_4$ dynamics. For example, compared to the wet and often snow−covered

high–latitude Arctic tundra ecosystems, there is no or little snow cover during the cold season in

the QTP alpine steppes. Jansson and Taş (2014) pointed out that relatively dry soils could



faciliate the oxidation of $CH_4$, since the increased number of gaps between soil particles in dry
soils enhances the diffusion of oxygen ($O_2$) and $CH_4$ molecules and promote aerobic respiration
of soil microorganisms (Wang et al., 2014; Song et al., 2015). Meanwhile, unfrozen or
capillary water found in cold–season permafrost soils ensures sufficient soil moisture for
microbial activities even in relatively drier and cold soils (Panikov and Dedysh, 2000; Rivkina et
al., 2004). In addition, many previous studies used static chambers in $CH_4$ measurements, and
may not include plant non–growing season (Wei et al., 2015a; Wang et al., 2014). Static
chambers could underestimate $CH_4$ uptake because of the additional chamber heating-induced
$CH_4$ emissions and frequent measurement gaps from overheating preventive shutdown
(Sturtevant et al., 2012).

Cryoturbation processes are of typical characteristics for the QTP permafrost (Wang et al.,

2008; Wang et al., 2000; Wu et al., 2010). Our work suggests that cryoturbation dynamics have
played a critical role in governing permafrost seasonal and diel $CH_4$ cycling. For instance, while
both spring_ and autumn_ are active seasons for the freeze-thaw dynamics of top soil layers and
share many similarities, they have opposite $CH_4$ processes–soils emit $CH_4$ during spring_ but
consume $CH_4$ during autumn_ (Figure 13: a and c). We hypothesize that the difference in the
cryoturbation process of the two seasons may have played a critical role in determining the
direction of $CH_4$ dynamics. In spring_, the active soil layer thaws from top to bottom (Jin et al.,
2000; Cao et al., 2017), and the permafrost table is very shallow (about 10 $\sim$ 45 cm) and is
generally water proof (Wu and Zhang, 2008; Song et al., 2015; Lin et al., 2015). The water
thawed during the day time would freeze again at night on the soil surface (Shi et al., 2006; Wu
and Zhang, 2010b). The thin-ice layer could stop atmospheric gases of $CH_4$ and $O_2$ from getting
into soils (Gazovic et al., 2010). During autumn_, however, soils are frozen from both top and



bottom and the permafrost table is deep (about 200 ～ 400 cm) (Wu and Zhang, 2010a), which
prevents the formation of a layer of thin ice during nighttime surface soil freezing. Instead, the
freezing of surface soil reduces soil liquid water content and creates fine gaps that allow $CH_4$ and
$O_2$ gases into deep soils (Mastepanov et al., 2008; Mastepanov et al., 2013; Zona et al., 2016).
Meanwhile, the temperature of deep soils still remains at a relatively high level and
methanotrophic bacteria there are still active at this high Tsoil. This could be one important
mechanism for autumn_ soil $CH_4$ consumption.

Furthermore, during the autumn_ thawing–freezing process, the vertical distribution of

clay, sandy soils, and soil organic layers was mixed like a multi–layer hamburger structure
(Figure 15), rather than forming a gradual change. Similarly, Tsoil, SWC, and soil microbial
activities also had this hamburger type of vertical distribution. As a result, layers of frozen and
thawed soils were not changing gradually, but appeared like a hamburger structure too. This
hamburger–like soil vertical structure trapped high concentration of soil water between the
frozen layers, which was therefore highly anaerobic and suitable for $CH_4$ production. Also
because of the hamburger–like structure, biogenic $CH_4$ between frozen layers could not escape in
autumn_, until when the top soil layer was completely frozen in winter_ and created frost cracks.
These explain the large burst of $CH_4$ emission in late autumn_ and early winter_ and may also
explain the constant weak $CH_4$ emission through the winter_ season, although methanogenic
bacteria may have stopped functioning in the low temperature of winter_.
**4.3 Impacts of Environmental, Permafrost, and Microbial Activities on $CH_4$ Fluxes**

Our results demonstrated the important roles of climate, cryoturbation dynamics, and soil

microbe activities in regulating the direction and amount of $CH_4$ exchanges between the
atmosphere and ecosystems in permafrost areas. This was also confirmed by the better





representation of seasonal $CH_4$ cycles by our new season division method based on soil microbes,
temperature, and permafrost dynamics rather than Tair or vegetation phenology. Here we further
discuss potential mechanisms of how environmental (including air and soil heat and water),
cryoturbation processes, and soil microbes control the production and absorption of $CH_4$.

First, it is noteworthy that both the strength and direction of correlations between $CH_4$

fluxes and SWC and Tsoil parameters changed with soil depths, particularly during spring_ and
autumn_ when active layer soils shifted between thawing and freezing regularly. The positive
and negative $CH_4$ flux correlations with Tsoil and SWC may suggest that the impacts of Tsoil
and SWC on $CH_4$ fluxes shall be treated as a holistic process rather than separate ones. For
instance, in autumn_, the correlation between $CH_4$ fluxes and Tsoil or SWC was positive at some
soil depths, but negative at some other depths, reaching the maximum at the depth of 80 cm.
Further *in situ* observations suggested that soil organic matter and soil microbe amount were also
at a very high level of this depth, highlighting the regulation of soil abiotic factors on $CH_4$
cycling may be well influenced by soil biotic activities. In addition, the holistic soil heat–water
process could also determine the concentration of soil inorganic ions, particularly during spring_
and autumn_, which were critical factors controlling the amount of soil unfrozen water; and soil
unfrozen water in winter may be important for maintaining soil microbial activities (Panikov and
Dedysh, 2000; Rivkina et al., 2004).

Tair and precipitation impact $CH_4$ fluxes indirectly through their influences on Tsoil and

SWC (Zhuang et al., 2004; Lecher et al., 2015). Such indirect influences may often be
characterized with time–lagged effects (Koven et al., 2011). For instance, post–drought rainfall
events in summer_ can first promote soil $CH_4$ consumption (summer_ of 2014). This is because
certain soil moisture is needed for methanogenic bacteria to function (Del et al., 2000; Luo et al.,





2012). Yet prolonged rainfall will eventually lead $CH_4$ fluxes changing from negative (soils
consume $CH_4$) to positive (soils emit $CH_4$) fluxes. After rainfall events, $CH_4$ flux gradually
turned negative again with the decrease of SWC. As a result of these time–lagged effects, the
correlation coefficient between $CH_4$ fluxes and precipitation often appears very low, although
still statistically significant.

Second, soil methanogenic and methanotrophic bacteria could co–exist with different

optimal niches (e.g., ranges of Tair / Tsoil and SWC; Zhuang et al., 2013; Lau et al., 2015; Wei
et al., 2015a). For example, $CH_4$ diel cycle in summer_ was found to have two strong
consumption peaks and one weak emission peak. The timing of these different peaks may well
reflect the different environmental requirements for the dominance of methanogens and
methanotrophic bacteria. Furthermore, methanogens may have a broader functional temperature
range than methanotrophic bacteria (Kolb, 2009; Lau et al., 2015; Yang et al., 2016). This is also
evident, for example, by the diel $CH_4$ cycle in autumn_ when $CH_4$ consumption was minimal at
both lowest and highest Tair.

The complex relationships between $CH_4$ fluxes and environmental factors make it a grand

challenge to predict the future of QTP $CH_4$ budget under changing climate. For instance, it has
been generally believed the ALT will increase under projected warming (Wu and Liu, 2004); and
the positive correlation between $CH_4$ fluxes and ALT found here suggests that the QTP
permafrost $CH_4$ sink may thus be weakened. However, the negative correlation between $CH_4$
flux and Tair may lead to a different conclusion. Incorporating our findings and high–resolution
data into mechanistic $CH_4$ models is therefore needed to enhance our capacity in predicting
future $CH_4$ budgets. Earth system models have been introduced to estimate $CH_4$ dynamics (Curry,
2007; Spahni et al., 2011; Bohn et al., 2015). For example, using a terrestrial ecosystem



modelling approach, Zhuang et al. (2004) estimated the average QTP permafrost $CH_4$ sink of -
0.08 g C $m^{-2}$ $yr^{-1}$, much smaller than our field–based $CH_4$ estimate. Current $CH_4$ models focus on
the regulation of $CH_4$ processes by temperature and SWC, and usually lack high–resolution data
for model parameterization (Bohn et al., 2015). Data interpolation and the use of average values
of certain environmental factors are normal practices in most models (Zhuang et al., 2004),
which may overlook the impacts of environmental variations on $CH_4$ dynamics. For example, at
Beilu'he, Tair of a typical summer day (e.g., July $6^{th}$, 2013) could vary between -6 °C and 28 °C,
a difference of 34 °C. The resulting diel mean temperature, 17 °C, is beyond the range of
methanotrophic bacteria's preferable temperature of 20～30 °C (Segers, 1998; Steinkamp et al.,
2001; Yang et al., 2016). Therefore, models using diel mean temperature as an input may
estimate the site as a net $CH_4$ sink. However, field observations show a source with a sink only
during a short period (8:30am～11:30 am) on July $6^{th}$, 2013 because the short–period of the sink
was offset by the source over the remaining 21 hours. Furthermore, half–hourly SWC was well
related with the waterproof role by permafrost during spring_ and autumn_ (Figure 6: a).
However, because of the shortage of high temporal resolution data, half–diel or diel mean SWC
data are often used in many previous studies (Zhu et al., 2004; Jiang et al., 2010; Wei et al.,
2015b), which could not correctly show the regulation of permafrost soil properties that are
critical for $CH_4$ dynamics. As another example, Tsoil of $0 - 50$ cm depth is one of the most
important factors related to $CH_4$ fluxes (Mastepanov et al., 2008). However, many studies used
Tair or re–analyzed deep Tsoil instead (Zhu et al., 2004; Bohn et al., 2015; Oh et al., 2016).
Because the active layer is not homogeneous but with different thermal conductivities during the
cryoturbation process, the use of Tair or deep Tsoil certainly brings in large uncertainties in $CH_4$
modelling. Future research needs to improve mechanistic understanding of $CH_4$ dynamics and


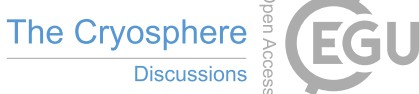

their biotic and abiotic control factors, and to conduct more high–resolution and long–term field
monitoring.

**5. Conclusions**

Our field data indicate there was a large $CH_4$ sink in the QTP permafrost area during the

recent years. The strength of this $CH_4$ sink is larger than previous studies in the region and many
high–latitude tundra ecosystems. This study highlights the complexity of environmental controls,
including soil heat–water processes, permafrost cryoturbation dynamics, and soil microbial
activities, on $CH_4$ cycling. This complexity implies that linear interpolation and extrapolation
from site-level studies could introduce large uncertainties in $CH_4$ flux estimation. Future
quantification of $CH_4$ dynamics in permafrost regions need to account for the effects of complex
environmental processes including cryoturbation, and the interaction between heat and water as
well as microbial activities. Our findings also highlight the importance of conducting more high–
resolution and long–term field monitoring in permafrost regions for better understanding and
modelling permafrost $CH_4$ cycling under a changing climate.

**Acknowledgements**

We would like to thank Yongzhi Liu, Jing Luo, Ji Chen, Guilong Wu, Wanan Zhu, Zhipeng Xiao,
and Chang Liao for their tremendous help in collecting field data over all these years. We also
want to pay tribute and gratitude to the late Xiaowen Cui for his contribution to our many field
adventures. This study was supported by the National Natural Science Foundation of China
(41501083), Opening Research Foundation of Key Laboratory of Land Surface Process and
Climate Change in Cold and Arid Regions, Chinese Academy of Sciences (LPCC201307), and
Opening Research Foundation of  Plateau Atmosphere and Environment Key Laboratory of
Sichuan Province (PAEKL − 2014 − C3). A. C. acknowledges the support from a Purdue



University Forestry and Natural Resources research scholarship. The data generated in this study
will be freely available on the Asia Flux regional network server
(https://db.cger.nies.go.jp/asiafluxdb/).

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

**Table 1**. Soil characteristics at the eddy covariance flux study site





| Soil depth cm | Soil type | Gravel content g kg$^{-1}$ | SOC g kg$^{-1}$ | Microbial Numbers $\times 10^4$ | pH | DBD g cm$^{-3}$ | SWC % | Total N $\times 10^3$ mg kg$^{-1}$ |
|---|---|---|---|---|---|---|---|---|
| 0 − 20 | clay | 22.3 | 2.8 | 3.44 | 8.7 | 1.75 | 18.26 | 0.87 |
| 20 − 50 | Silty clay | 12.6 | 1.7 | 3.82 | 8.4 | 1.73 | 11.52 | 1.02 |
| 50 − 120 | silt and fine sand | 3.4 | 1.3 | 3.67 | 8.4 | 1.72 | 12.57 | 1.18 |
| 120 − 160 | silt and fine sand | 2.8 | 26.4 | 5.44 | 5.1 | 1.68 | 24.69 | 2.46 |
| 160 − 200 | silt and fine sand | 1.6 | 13.6 | 4.39 | 6.8 | 1.68 | 22.45 | 2.03 |

**Note:** Gravel content diameter ≥ 0.5cm. SOC is soil organic content, DBD is dry bulk density,
and SWC is soil water content.













**Table 2.** Measurements of four seasons from 2012 to 2016

| | Spring_ | Summer_ | Autumn_ | Winter_ | Plant growing season | Plant non–growing season |
|---|---|---|---|---|---|---|
| | Period; Total days | Period; Total days | Period; Total days | Period; Total days | Period; Total days | Period; Total days |
| | Days | Days | Days | Days | Days | Days |
| 2012 | 50 – 142; 93 | 143 – 229; 87 | 230 – 323; 94 | 1 – 49, 324 – 366; 92 | 139 – 286; 148[a] | 1 – 138, 287 – 366; 218[a] |
| | | | | | 122 – 305; 184[b] | 1 – 121, 306 – 366; 182[b] |
| | | | | | 143 – 290; 148[c] | 1 – 142, 291 – 366; 218[c] |
| 2013 | 36 – 137; 102 | 138 – 224; 87 | 225 – 334; 110 | 1 – 35, 335 – 365; 66 | 139 – 287; 149[a] | 1 – 138, 288 – 365; 216[a] |
| | | | | | 121 – 304; 184[b] | 1 – 120, 305 – 365; 181[b] |
| | | | | | 127 – 297; 171[c] | 1 – 126, 298 – 365; 194[c] |
| 2014 | 49 – 127; 79 | 128 – 228; 101 | 229 – 309; 81 | 1 – 48, 310 – 365; 104 | 137 – 288; 152[a] | 1 – 136, 289 – 365; 213[a] |
| | | | | | 121 – 304; 184[b] | 1 – 120, 305 – 365; 181[b] |
| | | | | | 142 – 294; 153[c] | 1 – 141, 295 – 365; 212[c] |
| 2015 | 36 – 150; 115 | 151 – 224; 74 | 225 – 312; 88 | 1 – 35, 313 – 365; 88 | 145 – 288; 144[a] | 1 – 144, 289 – 365; 221[a] |
| | | | | | 121 – 304; 184[b] | 1 – 120, 305 – 365; 181[b] |
| | | | | | 136 – 295; 160[c] | 1 – 135, 296 – 365; 205[c] |
| 2016 | 47 – 161; 115 | 162 – 225; 64 | 226 – 299; 74 | 1 – 46, 300 – 366; 113 | 141 – 287; 147[a] | 1 – 140, 288 – 366; 219[a] |
| | | | | | 122 – 305; 183[b] | 1 – 120, 305 – 366; 182[b] |
| | | | | | 140 – 296; 157[c] | 1 – 139, 297 – 366; 209[c] |




**Note:** [a], based on vegetation cover and temperature change (VCT) (Lund et al., 2010; Tang and Arnone, 2013; Song et al., 2015); [b], based on Julian months (JMC) (Da et al., 2015); [c], based on vegetation phenology change (VPC). Spring_, Summer_, Autumn_, Winter_ are based on parameters of microbial activities, ALT variety coefficient and Tsoil (SMT).




**Table 3.** Correlation coefficients between CH$_4$ fluxes and environment factors on half–hour scales

| Environment Factors | CH4 Flux | | | | | | | | | |
| --- | --- | --- | --- | --- | --- | --- | --- | --- | --- | --- |
| | Spring_ | | Summer_ | | Fall_ | | Winter_ | | 2012 – 2016 | |
| | r | n | r | n | r | n | r | n | r | n |
| T$_{air}$ | 0.25** | 24144 | 0.14** | 19818 | -0.16** | 20959 | 0.32** | 22224 | -0.11** | 87145 |
| Wind Speed | 0.31** | 24144 | -0.04** | 19817 | -0.20** | 20959 | 0.32** | 22224 | 0.15** | 87144 |
| VPD | -0.33** | 18624 | -0.21** | 19263 | -0.09** | 16737 | -0.21 | 18000 | 0.26** | 69624 |
| Rn | 0.55** | 24143 | 0.09** | 19807 | -0.33** | 20913 | 0.51** | 22224 | 0.09** | 87087 |
| Albedo | 0.07** | 24144 | -0.01 | 19814 | -0.08** | 20913 | 0.10** | 22224 | 0.02** | 87095 |
| SHF of 5cm | 0.46** | 24144 | -0.08** | 19818 | -0.23** | 20913 | 0.43** | 22224 | 0.09** | 87099 |
| SHF of 15cm | 0.36** | 24144 | -0.15** | 19815 | -0.23** | 20913 | 0.33** | 22224 | 0.08** | 87096 |
| SWC of 10cm | -0.16** | 24144 | -0.14** | 19818 | -0.06** | 20959 | 0.00 | 22224 | -0.25** | 87145 |
| SWC of 20cm | -0.15** | 24144 | -0.13** | 19816 | -0.07** | 20959 | 0.11** | 22224 | -0.24** | 87143 |
| SWC of 40cm | -0.11** | 24144 | -0.02** | 19818 | 0.07** | 20959 | 0.06** | 22224 | -0.17** | 87145 |
| SWC of 80cm | | | -0.13** | 19818 | 0.06** | 20959 | | | | |
| SWC of 160cm | | | 0.04** | 19818 | -0.11** | 20959 | | | | |
| Precipitation | | | -0.02 | 16748 | 0.01$^b$ | 17888 | | | | |
| ALT | 0.73** | 23004 | 0.23** | 19823 | 0.73** | 21454 | | | 0.43** | 64281 |


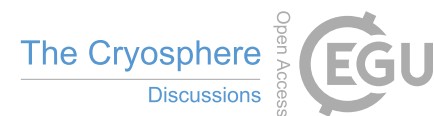

|  | r | n | r | n | r | n | r | n | r | n |
|---|---|---|---|---|---|---|---|---|---|---|
| ΔI | 0.77** | 100 | 0.57** | 83 | 0.46** | 89 | 0.23 | 93 | 0.49** | 365 |
| ΔII | 0.31** | 100 | 0.66** | 83 | 0.78** | 89 | 0.19 | 93 | 0.52** | 365 |
| $T_{soil}$ of 0 cm | -0.06* | 23004 | 0.13** | 19823 | 0.07** | 20366 | 0.13** | 21711 | 0.11** | 84904 |
| $T_{soil}$ of 5 cm | 0.15** | 24144 | 0.15** | 19808 | -0.13** | 21454 | 0.27** | 22224 | 0.24** | 87630 |
| $T_{soil}$ of 10 cm | -0.03** | 24144 | 0.12** | 19808 | 0.08** | 21454 | 0.16** | 22224 | 0.13** | 87630 |
| $T_{soil}$ of 20 cm | -0.14** | 24144 | 0.08** | 19808 | 0.02** | 21454 | 0.06** | 22224 | -0.09** | 87630 |
| $T_{soil}$ of 30 cm | -0.13* | 23004 | 0.06** | 19823 | -0.02** | 20366 | 0.07** | 21711 | -0.08** | 84904 |
| $T_{soil}$ of 40 cm | 0.14** | 24144 | 0.05** | 19808 | -0.01^b | 21454 | 0.06** | 22224 | 0.11* | 87630 |
| $T_{soil}$ of 50 cm |  |  | 0.04** | 19823 | -0.05** | 20366 |  |  |  |  |
| $T_{soil}$ of 70 cm |  |  | 0.07** | 19823 | -0.05** | 20366 |  |  |  |  |
| $T_{soil}$ of 80 cm |  |  | 0.05** | 19808 | 0.04** | 21454 |  |  |  |  |
| $T_{soil}$ of 100 cm |  |  | 0.10** | 19823 | -0.05** | 21454 |  |  |  |  |
| $T_{soil}$ of 150 cm |  |  | 0.09** | 19823 | -0.04** | 20366 |  |  |  |  |
| $T_{soil}$ of 160 cm |  |  | 0.10** | 19808 | 0.01** | 21454 |  |  |  |  |
| $T_{soil}$ of 200 cm |  |  | 0.02** | 19823 | -0.02** | 20366 |  |  |  |  |

**Note:** ** means $p<0.01$, * means $p<0.05$; r values for the relationship between $CH_4$ flux and environment factors. Tair means air temperature of 3 m above the ground surface. VPD is vapor pressure deficit, NR is net radiation, and SWC is soil water content, ALT is active layer thickness, which fitted through the depth of soil 0 °C in Surfer 8.0., and the data is removed of meaningless in winter. Tsoil is the temperature of the soil. In spring_ and



winter_, precipitation data are too sparse for statistical analysis. $\Delta I$ is the soil $0-25cm$ archaeal methanogens gene expression, and $\Delta II$ is the soil $0-$
25 cm methanotrophic gene expression. The coefficients (r) between $CH_4$ flux and $\Delta I$, $\Delta II$ are obtained using the synchronous $CH_4$ fluxes averaged
for 5 days.













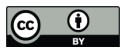

**Table 4.** Principal components analysis (PCA) of the environmental factors.

| Component | Spring PC1 | PC2 | PC3 | PC4 | Summer PC1 | PC2 | PC3 | PC4 | autumn PC1 | PC2 | PC3 | PC4 | Winter PC1 | PC2 | PC3 | PC4 |
|---|---|---|---|---|---|---|---|---|---|---|---|---|---|---|---|---|
| wind speed | -0.03 | 0.51 | 0.65 | -0.46 | 0.02 | 0.37 | 0.38 | -0.13 | -0.04 | 0.44 | 0.59 | 0.67 | 0.27 | 0.45 | -0.11 | -0.27 |
| Tair | 0.38 | 0.29 | -0.05 | -0.11 | 0.42 | 0.22 | -0.03 | 0.02 | 0.36 | 0.21 | 0.08 | -0.06 | 0.48 | 0.12 | -0.02 | 0.01 |
| VPD | 0.34 | -0.27 | 0.40 | 0.15 | 0.17 | 0.46 | -0.22 | 0.09 | 0.34 | -0.15 | 0.17 | -0.07 | 0.14 | -0.15 | 0.95 | -0.22 |
| Rn | 0.16 | 0.49 | 0.00 | 0.76 | -0.01 | 0.07 | 0.58 | 0.11 | 0.12 | 0.54 | -0.43 | -0.07 | 0.26 | 0.47 | -0.01 | -0.49 |
| SHF of 15cm | 0.24 | 0.49 | -0.30 | -0.09 | 0.25 | 0.53 | -0.09 | 0.01 | 0.15 | 0.59 | -0.23 | -0.15 | 0.36 | 0.37 | 0.14 | 0.58 |
| ALT | 0.22 | -0.40 | 0.40 | 0.27 | 0.32 | -0.53 | -0.05 | 0.02 | 0.29 | 0.49 | 0.70 | 0.25 |  |  |  |  |
| $\Delta I$ | 0.49 | -0.22 | 0.01 | -0.08 | 0.50 | -0.16 | 0.02 | -0.16 | 0.29 | 0.31 | 0.24 | -0.51 | 0.52 | 0.05 | 0.07 | -0.03 |
| SWC of 10 − 20cm |  |  |  |  |  |  |  |  |  |  |  |  | -0.31 | 0.45 | 0.22 | 0.47 |
| SWC of 10 − 40cm | 0.33 | -0.20 | 0.50 | 0.25 | -0.16 | 0.15 | -0.16 | 0.73 | 0.28 | -0.18 | -0.41 | 0.53 |  |  |  |  |
| SWC of 50 − 160cm |  |  |  |  | 0.23 | -0.20 | -0.16 | 0.55 | 0.31 | -0.17 | -0.32 | 0.41 |  |  |  |  |
| Precipitation |  |  |  |  | 0.03 | -0.04 | 0.63 | 0.35 |  |  |  |  |  |  |  |  |
| Tsoil of 0 cm | 0.43 | -0.07 | -0.20 | -0.27 | 0.43 | 0.08 | 0.08 | -0.07 | 0.37 | 0.07 | 0.19 | -0.16 | 0.43 | -0.35 | -0.15 | 0.09 |
| Tsoil of 5 − 20 cm | 0.44 | -0.01 | -0.17 | -0.16 |  |  |  |  |  |  |  |  | 0.45 | -0.28 | 0.00 | 0.28 |
| Tsoil of 5 − 40 cm |  |  |  |  | 0.46 | -0.05 | 0.04 | -0.03 | 0.38 | 0.02 | 0.18 | -0.17 |  |  |  |  |
| Tsoil of 30 − 50cm | 0.40 | -0.23 | -0.08 | -0.04 |  |  |  |  |  |  |  |  |  |  |  |  |
| Tsoil of 50 − 80cm |  |  |  |  | 0.37 | -0.36 | 0.00 | 0.01 | 0.37 | -0.11 | 0.19 | -0.14 |  |  |  |  |
| Tsoil of 100 − 200cm |  |  |  |  | 0.33 | -0.34 | 0.01 | -0.01 | 0.36 | -0.14 | 0.08 | 0.00 |  |  |  |  |
| Percent of variance | 0.63 | 0.23 | 0.08 | 0.04 | 0.70 | 0.18 | 0.07 | 0.02 | 0.69 | 0.17 | 0.08 | 0.04 | 0.75 | 0.21 | 0.02 | 0.01 |
| Cumulative | 0.63 | 0.86 | 0.94 | 0.98 | 0.70 | 0.88 | 0.95 | 0.97 | 0.69 | 0.86 | 0.94 | 0.98 | 0.75 | 0.96 | 0.98 | 0.99 |

**Note:** PC means principal component. Before PCA, SWC was divided for three parts, 10 – 20 cm, 10 – 20 cm, 10 – 40 cm, and 50 – 160 cm according to
collinearity test in four seasons. Tsoil was divided for six parts of Tsoil of 0 cm, Tsoil of 5 – 20 cm, Tsoil of 5 – 40 cm, Tsoil of 30 – 50 cm, Tsoil of
50 – 80 cm, and Tsoil of 60 – 200 cm according to collinearity test in different seasons.






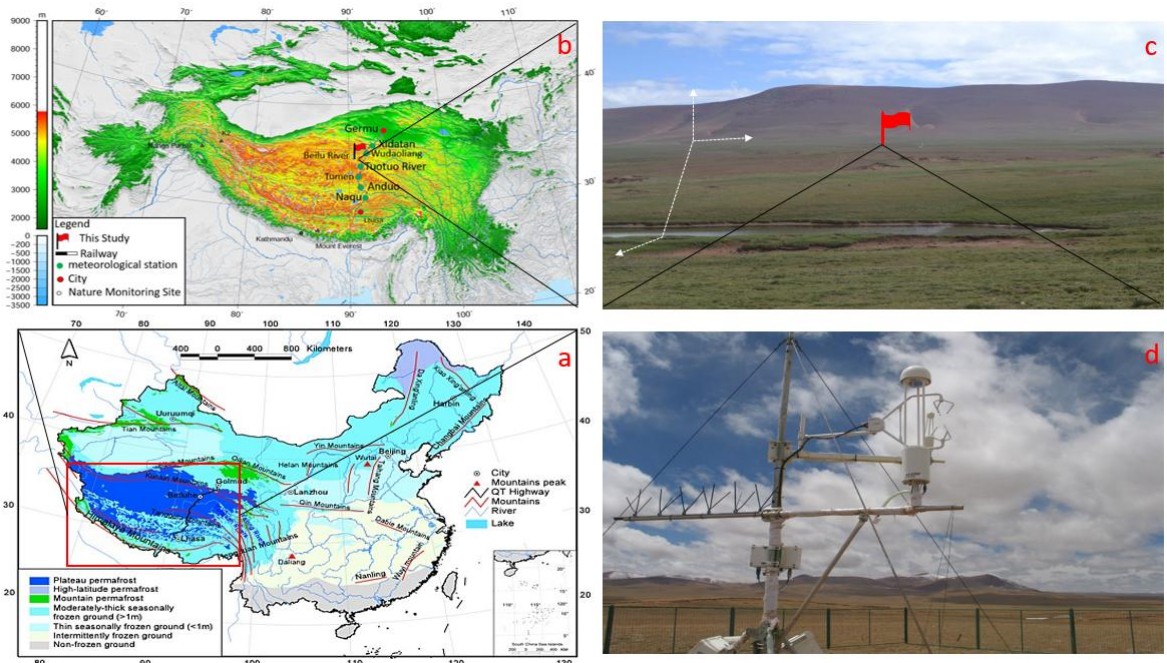


**Figure 1.** Geographic location of the study site: (a) is a map of China's permafrost distribution,

and the red box marks the approximate location of the Qinghai–Tibet Plateau; (b) shows the

study site location and meteorological stations along the Qinghai–Tibet railway; (c) is the photo

showing the study site's topography and physiognomic. The small red flag in (c) is the eddy

covariance tower location; (d) is the close–up shot of the LI–7700 for methane measurement.

*Map boundary and location are approximate. Geographic features and the names do not imply*

*any official endorsement or recognition.*





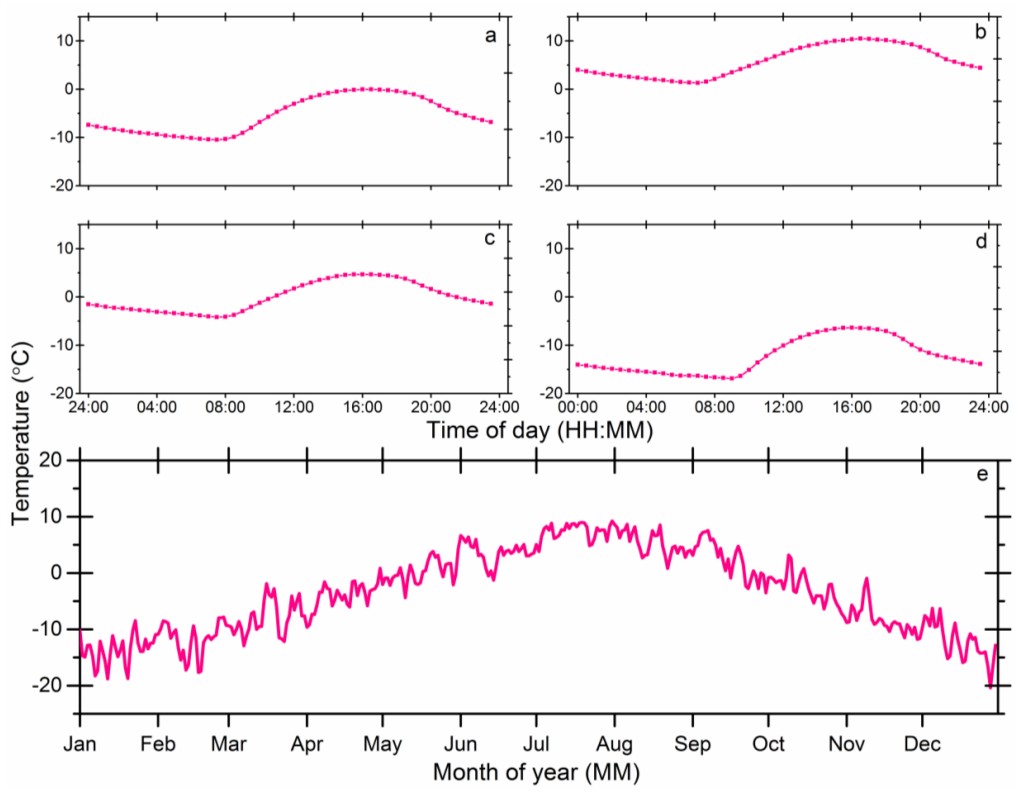

**Figure 2.** Air temperature (Tair) of 3 meters above the ground surface: (a), (b), (c), and (d) are

half–hour scale mean values in spring, summer, autumn, and winter, respectively; (e) shows

diel–scale mean values from 2012 to 2016.



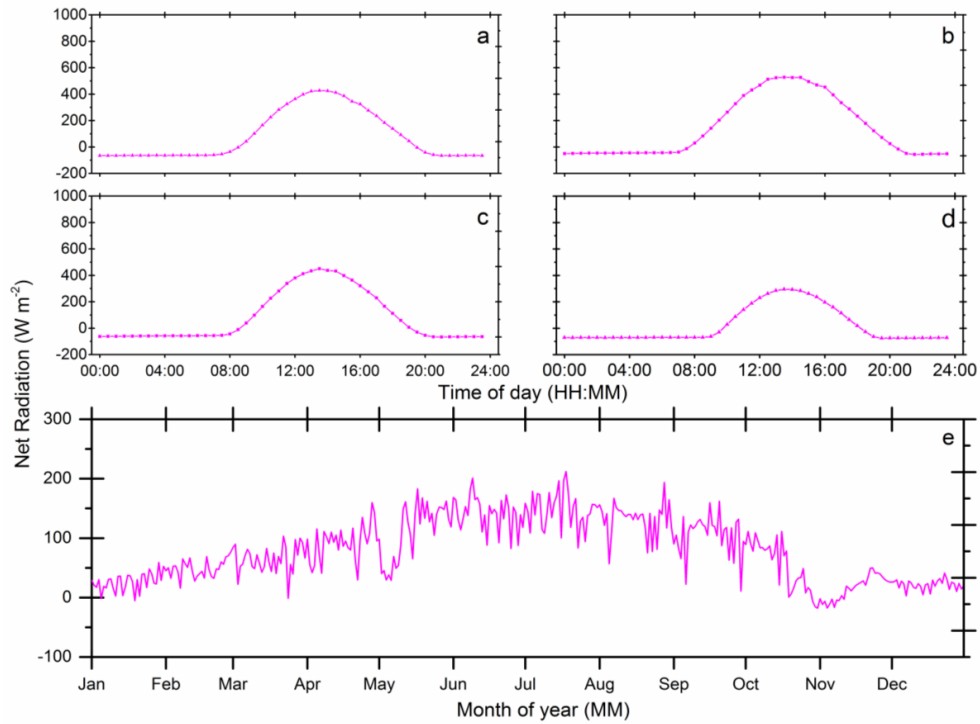

**Figure 3.** Net radiation (Rn) of 3 meters above the ground surface: (a), (b), (c), and (d) are half–

hour scale mean values in spring, summer, autumn, and winter, respectively; (e) shows diel–

scale mean values from 2012 to 2016.



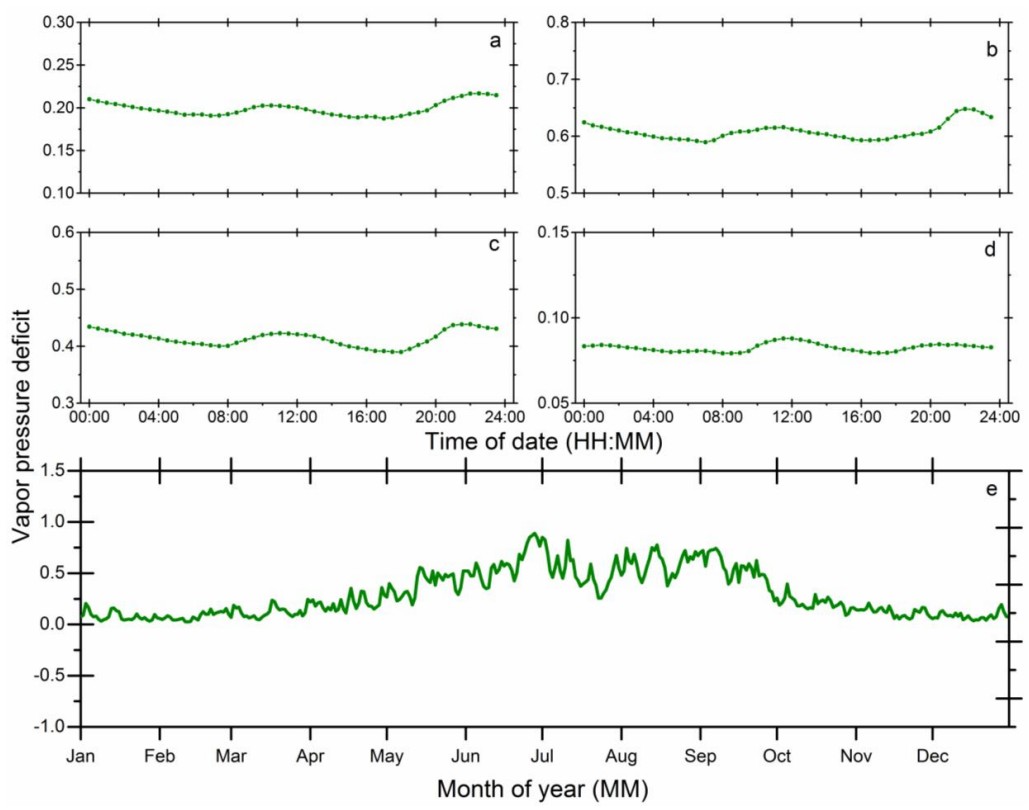

**Figure 4.** Vapor pressure deficit (VPD) of 3 meters above the ground surface: (a), (b), (c), and (d)

are half–hour scale mean values in spring, summer, autumn, and winter, respectively; (e) shows

diel–scale mean values from 2012 to 2016.





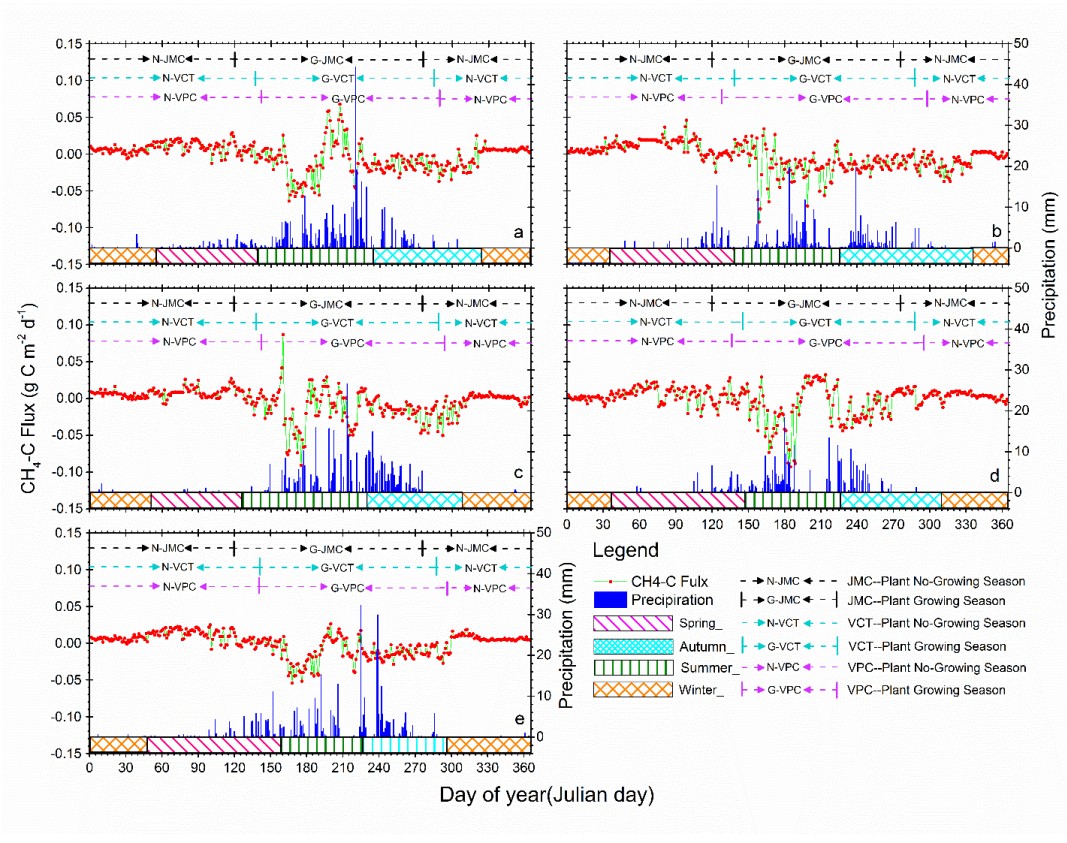

**Figure 5.** Annual patterns of diel methane (CH₄) flux and precipitation variations from 2012 to

2016. Positive values indicate CH₄ release and negative values indicate CH₄ uptake by

ecosystems. Red dots and light green lines are CH₄–C flux variation, and the deep blue

histograms show diel precipitation accumulation. Pink, olive, cyan, and orange blocks mean

spring, summer, autumn, and winter seasons according to our new method of SMT (see

Methods), respectively. Black, cyan, and pink dotted lines with bars separating the plant growing

from non–growing seasons stand for seasons by the method JMC, VCT, and VPC, respectively.

Details about the methods JMC, VCT, and VPC can be found in Text part 3.2.



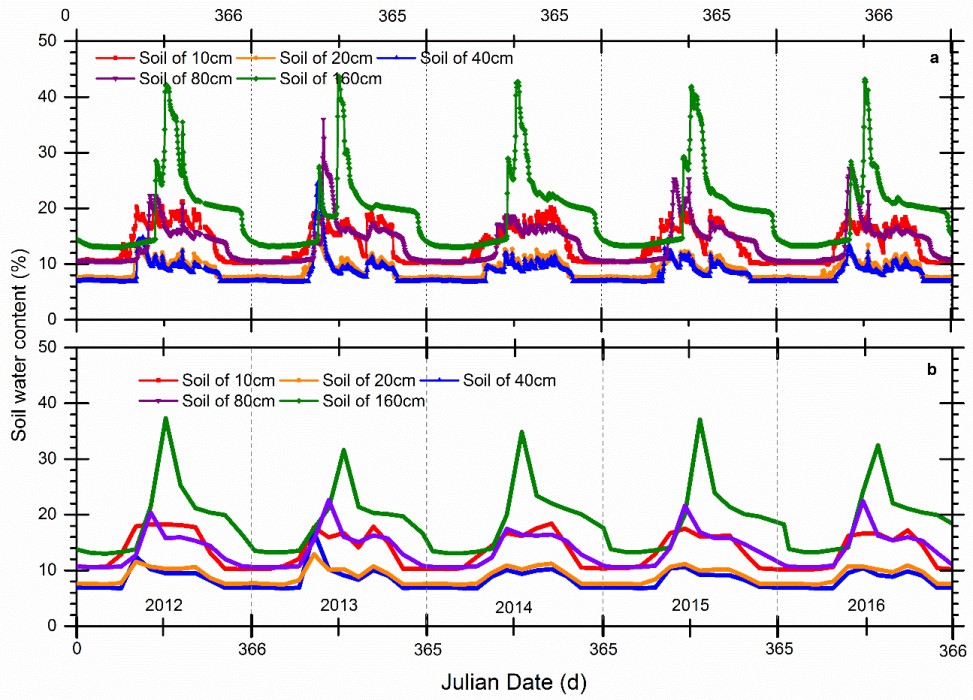

830

**Figure 6.** Comparison between soil water content (SWC) of two different time resolutions from

2012 to 2016, (a) is the half–hour scale SWC at soil depths of 10 cm, 20 cm, 40 cm, 80 cm, and

160 cm; and (b) is the 4–hour mean SWC for the same depths.

834





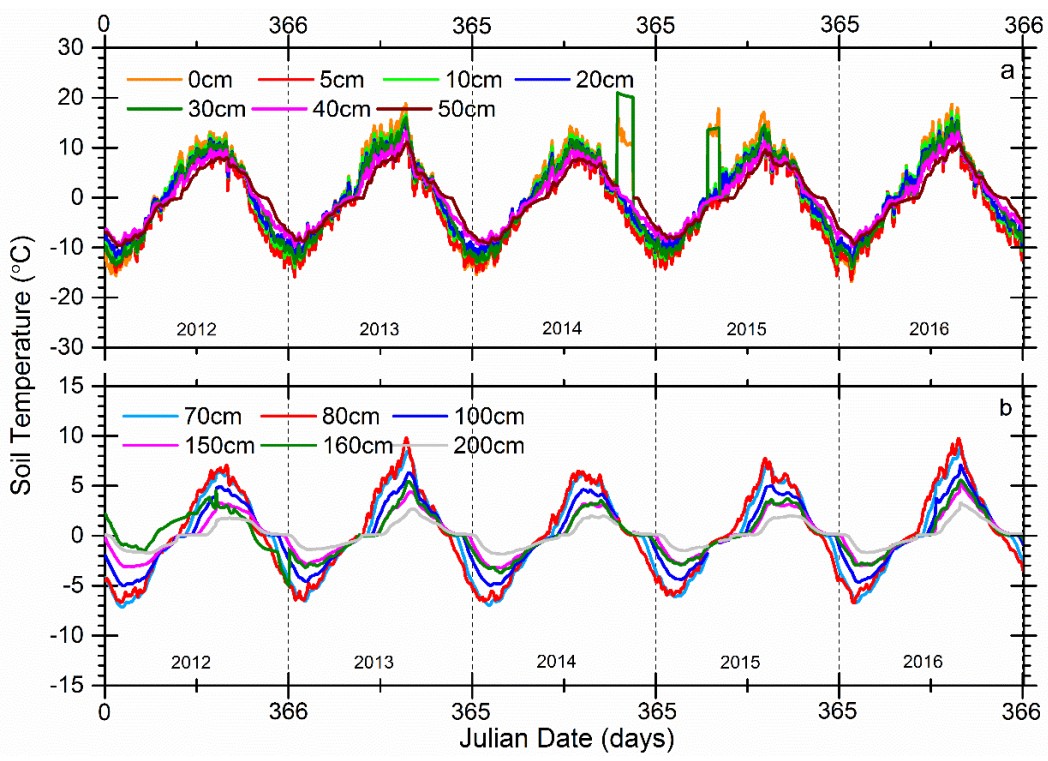

**Figure 7.** Half–hour scale of $0 - 200$ cm soil temperature (Tsoil) variations from 2012 to 2016,

(a) is for soil depths of 0 cm, 5 cm, 10 cm, 20 cm, 30 cm, 40 cm, 50 cm, (b) is for soil depth of

70 cm, 80 cm, 100 cm, 150 cm, 160 cm, and 200 cm.





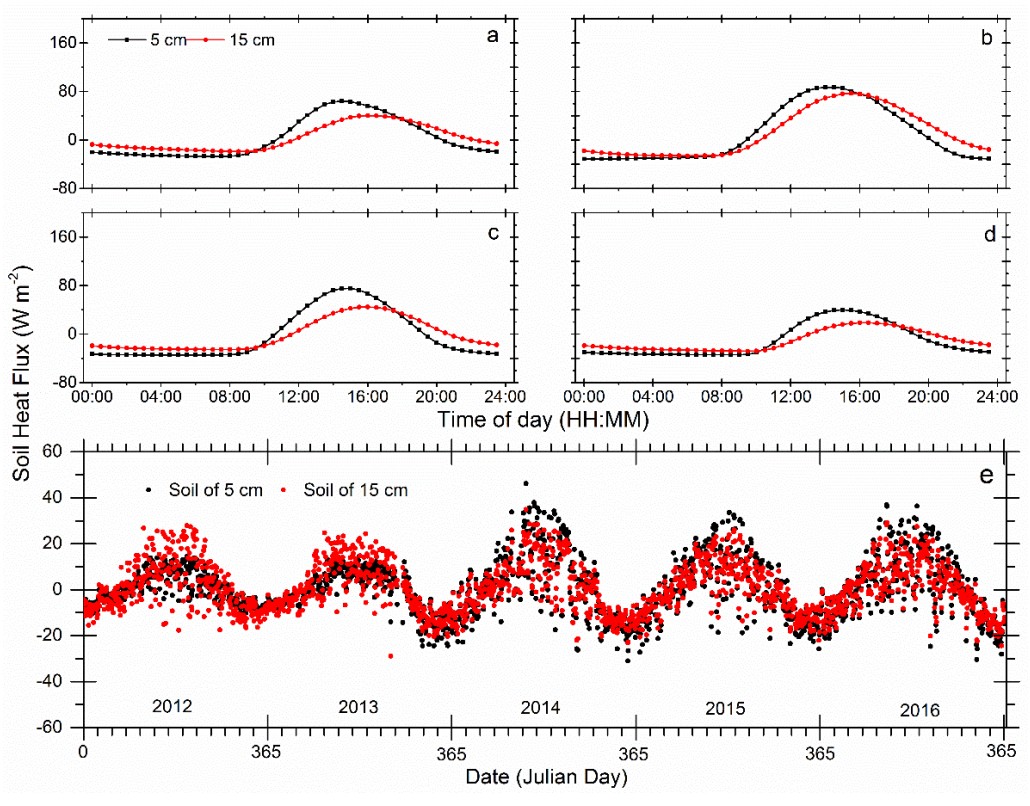


**Figure 8.** Soil heat flux (SHF) at depth of 5 cm and 15 cm: (a), (b), (c), and (d) are half–hour

scale mean values in spring, summer, autumn, and winter, respectively; (e) shows diel–scale
mean values from 2012 to 2016.



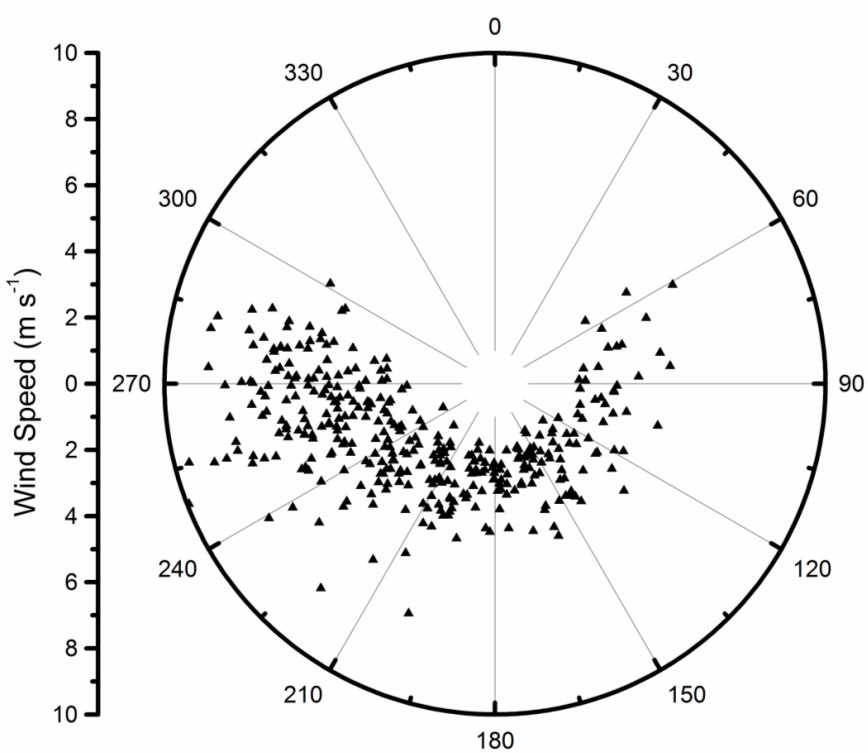

**Figure 9.** Diel mean of wind speed and direction between 2012 and 2016. All data are presented

as mean values with standard deviations (mean ± standard deviation).

.



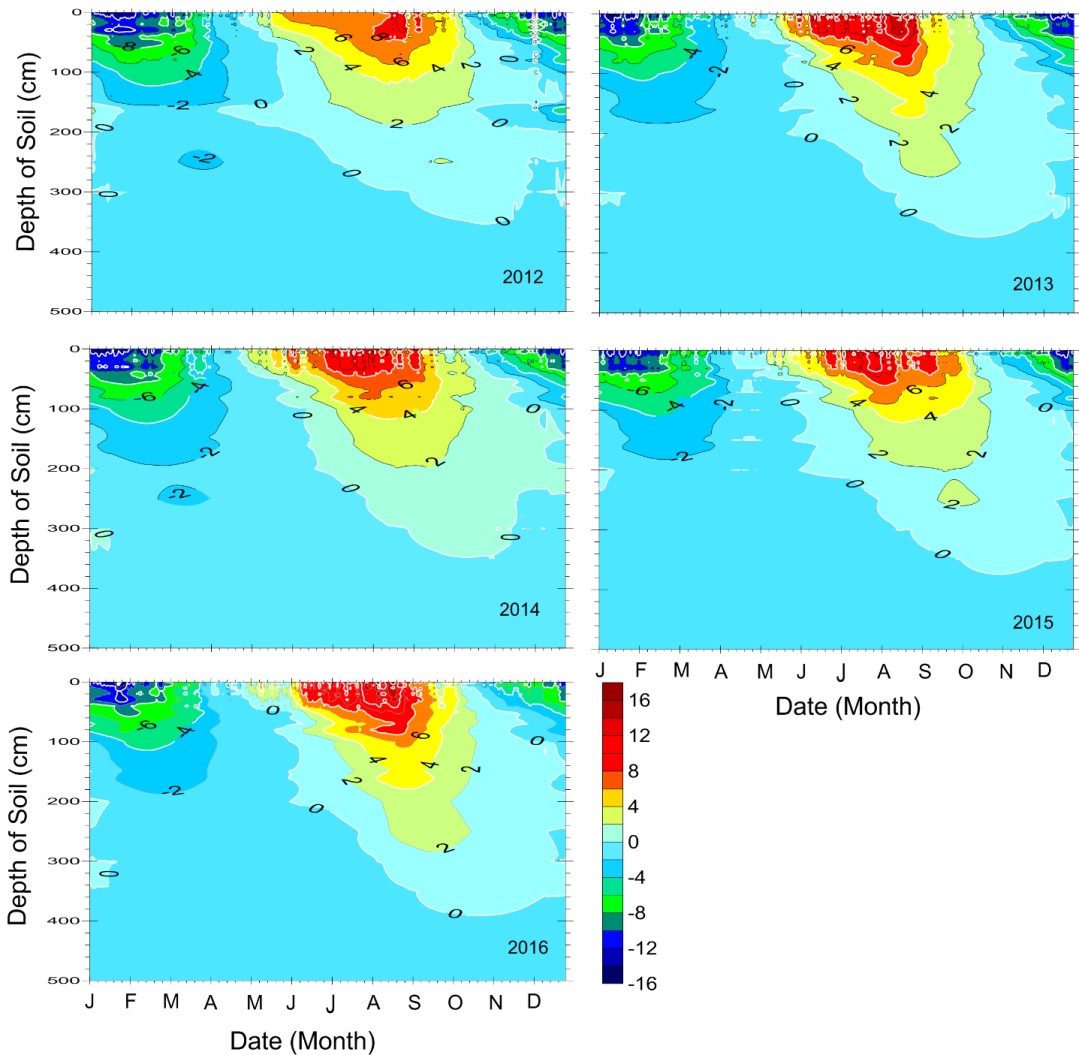

**Figure 10.** Characteristics of the seasonal freezing and thawing processes of the active layer for

years: 2012, 2013, 2014, 2015, and 2016. Different colors represent the soil temperature

gradients from -16 ℃ to 20 ℃. The depth of 0 ℃ represent the active layer thickness (ALT).





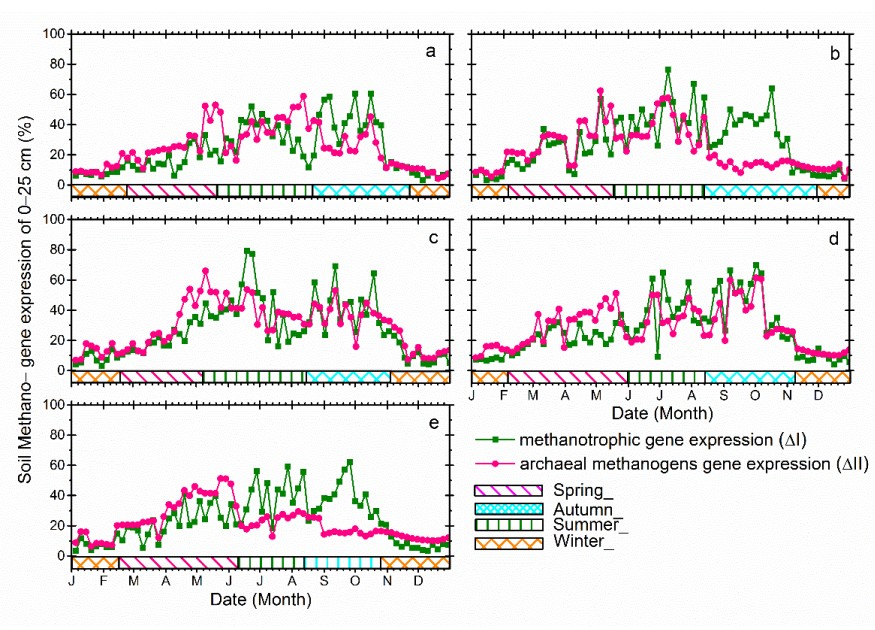


**Figure 11.** Annual patterns of soil methanogen−gene expression of $0 − 25$ cm soil depth for

years: (a) 2012, (b) 2013, (c) 2014, (d) 2015, and (e) 2016.






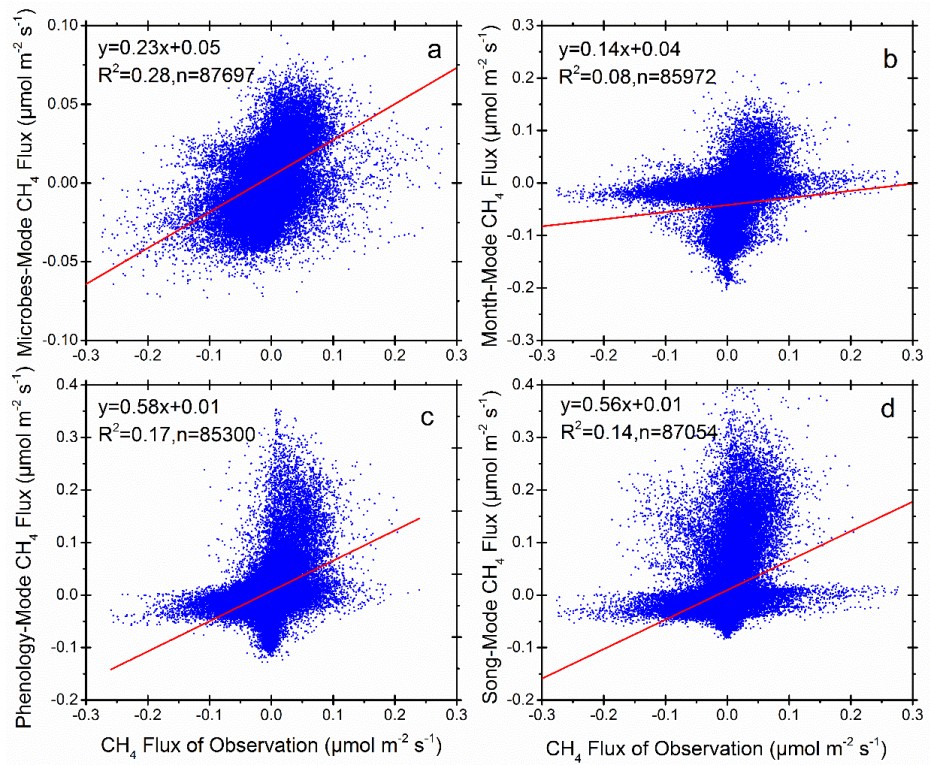

**Figure 12.** Regression comparison between observation and modeled methane fluxes with four

different seasonal definitions and classification models. Panels (a), (b), (c), and (d) are for the

SMT, JMC, VCT, and VPC methods, respectively.





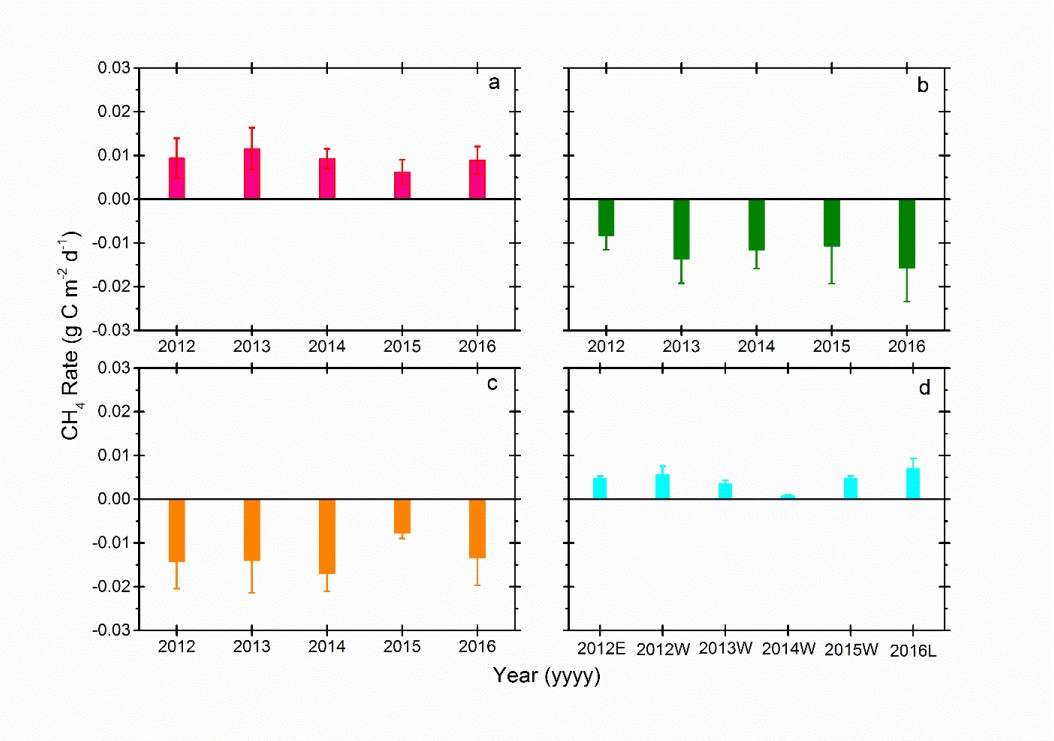

**Figure 13.** Seasonal $CH_4$ rate mean value from 2012 to 2016: (a) is spring, (b) is summer, (c) is

autumn, and (d) is winter. In the (c), 2012E is started from January 1[st], 2012 and ended on

February 17[th], 2012; 2012W is started from 19[th] November, 2012 to 4[th] February, 2013; 2013W

is started from 1[st] December, 2013 to 17[th] February, 2014; 2014W is started from 6[th] November,

2014 to 4[th] February, 2015; 2015W is started from 9[th] November, 2015 to 15[th] February, 2016;

2016L is started from October 26[th], 2016 and ended on December 31[st], 2016. All data are

presented as mean values with standard deviations (mean ± standard deviation).



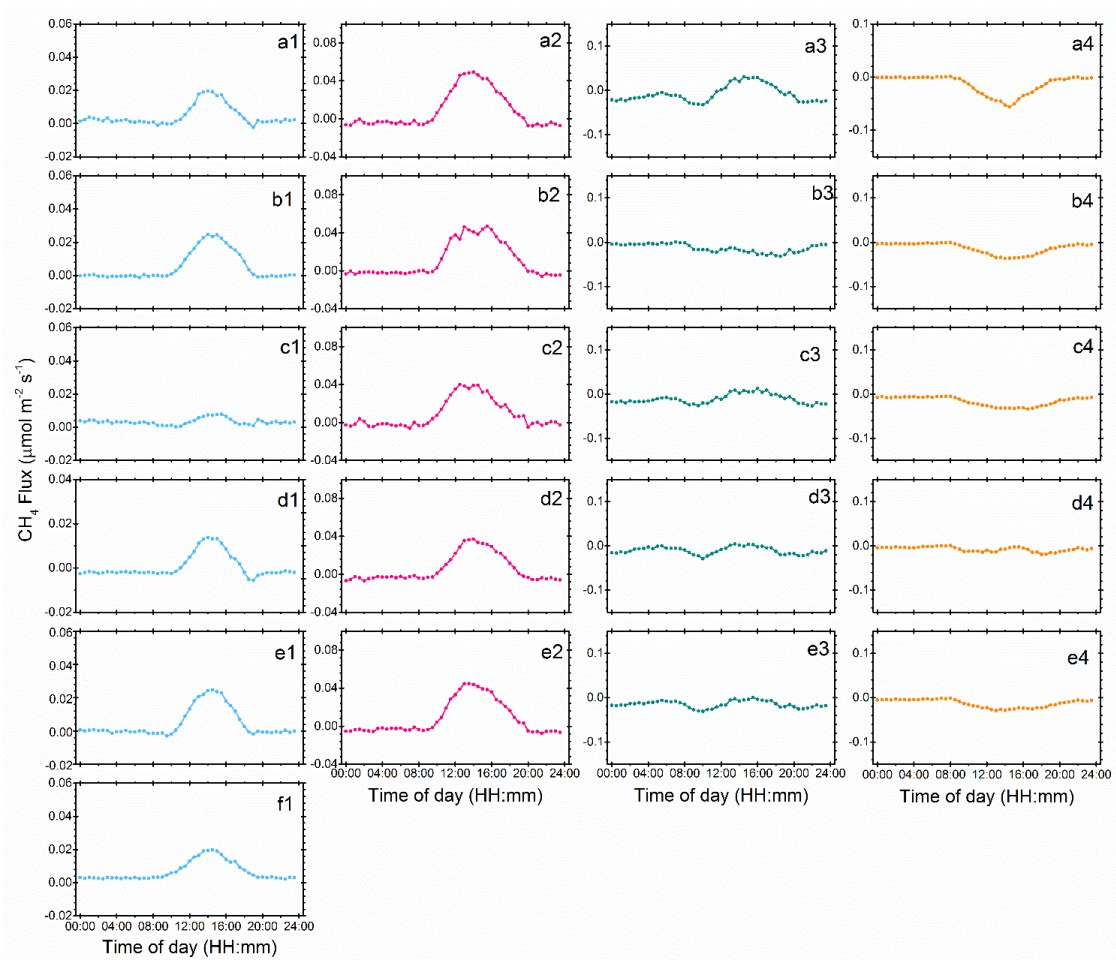

**Figure 14.** Diel CH$_4$ fluxes from 2012 to 2016 for different seasons. Pink, green, orange and

blue, represent spring, summer, autumn, and winter, respectively; (a1), (a2), (a3), and (a4) are for

2012; (b1), (b2), and (b3), and (b4) are for 2013; (c1), (c2), (c3) and (c4) are for 2014; (d1), (d2),

(d3), and (d4) are for 2015; (e1), (e2), (e3), (e4) and (f1) are for 2016.





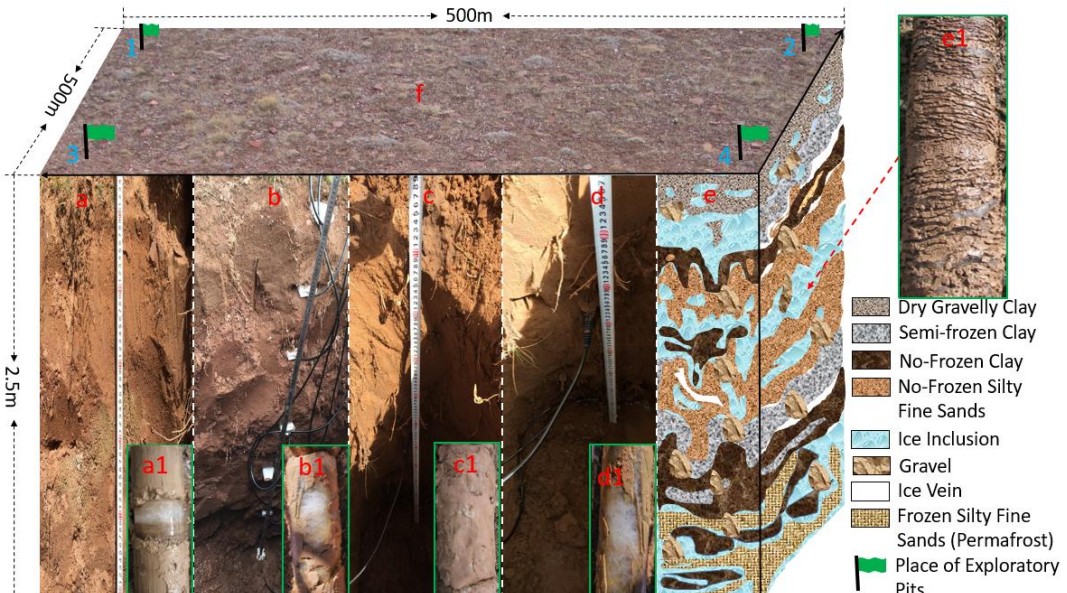

888

**Figure 15.** Location of exploratory pits and drillings in this study in autumn: (f) is photo of a

typical ground surface (October 16th, 2014). Green flags represent the location for soil survey by

test pitting and drilling. (a), (b), (c), and (d) are test pitting sections for active layer $0-250$ cm

depths soil water content and temperature measured in eddy covariance North (1), South

(2), East (3), and West (4) corners, respectively. (a1), (b1), (c1), and (d1) are drilling cores, with

clear ice (white) in (a1), (b1), and (d1), but not in (c1); (e) provides an illustration that combines

results from drillings, test pitting and multi–channel ground–penetrating radar (Malå Geoscience,

Sweden) for active layer variations in permafrost area during the autumn season; and (e1) is a

core sample of the same drilling (October 16th, 2014).

898

899