# Peer review of "Consumption of atmospheric methane by the Qinghai–Tibetan"

_The Cryosphere, 2017_

## Referee Comment (RC1) · Anonymous Referee #1 · 4 Jan 2018

Yun et al. present in this paper eddy covariance based observations of land-atmosphere methane exchanges in concert with environmental data, such as climate and soil temperature or water content. The data shows that the site at the Tibet plateau is a net annual methane sink, which is an important finding for better constraining bottom-up estimates of the methane balance. Most semi-empirical models so far do not allow any methane sink but assume a methane source of soils. Interestingly, this ecosystem, if uniform in soil properties and vegetation, seems to act as methane source during winter and spring while acing as a methane sink in summer and autumn.

These findings are interesting and important, and the paper is in general well written.

[Figure]

Still, I do have some questions and comments that should be addressed thoroughly prior to any publication.

i) This is a good introduction into the topic and knowledge gap. However, I request to state the research questions addressed here more precisely. I can see at least three questions: - What is the long-term annual methane budget of the study site? - What is the seasonal methane budget of the study site? - Which environmental factors control the seasonal methane budget and why? - Is a classical vegetation productivity based definition of growing season useful for defining the methane flux seasonality? The manuscript tries to address all these questions but it is useful to state them precisely in the introduction, and then they can be addressed with respective methods and presentation of results.

ii) Cryoturbation is a collective term for many soil transport processes based on freezing and thawing that lead to a subduction of topsoil horizons down to the transition zone. This term is wrongly used in this manuscript. I think what the authors mean is freezing and thawing instead. Please, correct this.

iii) In addition to eddy covariance based observations of methane fluxes, this study presents a lot of observations of environmental factors, such as climate and soil properties. This is really interesting because this allows addressing the question on why do we see this strong seasonality with methane sources (summer, autumn) and sinks (winter, spring). However, the data presented does not explain these seasonal differences. The attempts of explanation in the discussion section with several hypotheses are important but please include into this discussion how these hypotheses could have been proven by your data or which other measurements were required.

In general, the temporal differences in eddy cov CH4 time series could have been due to either temporal differences in soil processes or spatial differences of the footprint. I strongly suggest to first rule out the latter case before discussing all kind of soil processes leading to the seasonality of methane fluxes: Are there more wet or dry soil

areas in the footprint and do we see methane flux dynamics due to changes in the footprint? Please, analyze wind direction and wind speed together with methane fluxes in Fig. 5. Also, the main wind direction can be displayed in color scale in fig 9.

One more idea that could be tested is the importance of vegetation activity for an oxygen flux into the soil. You could analyze your GPP data from the tower in concert with methane fluxes to prove this hypothesis.

Minor comments - I would place fig 13 and 14 directly after fig 5. Fig 14 should have the same order of seasons than fig 13 and a uniform y-axis scale. - Most of section 3.2 should be part of the methods section.

---

## Referee Comment (RC2) · Anonymous Referee #2 · 1 Feb 2018

This paper entitled "Consumption of atmospheric methane by the Qinghai-Tibetan Plateau alpine steppe ecosystem" describes a study of methane dynamics determined with a rich, multi-year microbial and eddie-covariance data set. The authors observed an interesting shift in the ecosystem from a CH4 source to a sink over the season and propose a new seasonal separation based on soil and microbial conditions rather than air temperature. The modeling effort was not terribly successful (only describing a small portion of the observed variation), but given the high temporal frequency and multi-year nature of the data, this seems like a very compelling contribution to this journal.

My main two critiques are about the paper's structure and number of figures. On the first point, there are many grammatical errors that distract from the message of the paper. Starting from the first lines of the abstract through the end of the paper, a thorough, line-by-line treatment is needed. More generally, the paper would greatly benefit from a thorough revision at the paragraph and section levels. Making sure there are clear topic sentences for each paragraph and that each section has a logical progression would help readers appreciate the importance of these findings. On the second point, there are many figures that are better suited for the supplementary information. Currently, including the background meteorological figures before getting to the response variable of interest (CH4 flux) reduces the focus and punch of the findings. Focusing on a few key figures (for example 5, and 11-14) would improve the paper.

---

## Author Comment (AC1) · 2 Jun 2018

Dear editor, We thank you for handling our manuscript [tc-2017-264], and thank you for your valuable and constructive comments and suggestions. Following those comments and suggestions, we have carefully revised the manuscript; and we are pleased to submit the revised manuscript for potential publication in The Cryosphere. In this revised manuscript, we have made the following major changes. 1. We have included explicit statements on research goals in the revised manuscript. 2. We strengthened the connection between data and mechanistic explanations, in which we showed how several hypotheses about the mechanisms behind our observed CH4 patterns can be proven

or tested with field measured environment data. 3. We re-organized all the figures by only keep some significant figures in main text and putting other non-essential figures into supplementary materials. 4. We enhanced the writing by ensuring that each paragraph has a topic sentence, and the flow of narratives progresses naturally. We also had the entire manuscript checked for grammatical errors by a native English speaker. We have also addressed all the other questions and comments raised by you. Detailed responses to each question and comment can be found in the responses to reviewer comments (see supplement files). We hope this revised manuscript will satisfy you.

Thank you again for your time and efforts on our manuscript. We look forward to hearing from you.

Yours sincerely,

Hanbo Yuns

Please also note the supplement to this comment:
https://www.the-cryosphere-discuss.net/tc-2017-264/tc-2017-264-AC1-supplement.zip

---

## Author Comment (AC2) · 2 Jun 2018

Dear reviewer, We thank you for handling our manuscript [tc-2017-264], and thank you very much foryourvaluable and constructive comments and suggestions. Following those comments and suggestions, we have carefully revised the manuscript; and we are pleased to submit the revised manuscript for potential publication in The Cryosphere. In this revised manuscript, we have made the following major changes. 1. We re-organized all the figures by only keep some significant figures in main text and putting other non-essential figures into supplementary materials. 2. We enhanced the writing by ensuring that each paragraph has a topic sentence, and the flow of narra-

tives progresses naturally. We also had the entire manuscript checked for grammatical errors by a native English speaker. We have also addressed all the other questions and comments raised by the other reviewer. Detailed responses to each question and comment can be found in the responses to reviewer comments(see supplement files). We hope this revised manuscript will satisfy you.

Thank you again for your time and efforts on our manuscript. We look forward to hearing from you.

Yours sincerely,

Hanbo Yun

Please also note the supplement to this comment:
https://www.the-cryosphere-discuss.net/tc-2017-264/tc-2017-264-AC2-supplement.zip

---

## Author Response (AR1)

**Responses to Reviewers**

Responses to reviewer#1

**General Comments**

Yun et al. present in this paper eddy covariance based observations of landatmosphere methane exchanges in concert with environmental data, such as climate and soil temperature or water content. The data shows that the site at the Tibet plateau is a net annual methane sink, which is an important finding for better constraining bottom-up estimates of the methane balance. Most semi-empirical models so far do not allow any methane sink but assume a methane source of soils. Interestingly, this ecosystem, if uniform in soil properties and vegetation, seems to act as methane source during winter and spring while acing as a methane sink in summer and autumn. These findings are interesting and important, and the paper is in general well written. Still, I do have some questions and comments that should be addressed thoroughly prior to any publication.

*Response:* Thank you very much for your insightful comments and great suggestions, which are tremendously helpful for us to improve the manuscript. Following your comments, we have revised the manuscript. Detailed responses to each of your comment and suggestion can be found in the following point-to-point responses.

1. i): This is a good introduction into the topic and knowledge gap. However, I request to state the research questions addressed here more precisely. I can see at least three questions: - What is the long-term annual methane budget of the study site? - What is the seasonal methane budget of the study site? - Which environmental factors control the seasonal methane budget and why? -Is a classical vegetation productivity based definition of growing season useful for defining the methane flux seasonality? The manuscript tries to address all these questions but it is useful to state them precisely in the introduction, and then they can be addressed with respective methods and presentation of results.

**Response:** Thank you for the suggestion of clearly stating the research questions in the Introduction. Following these suggestions, we have included explicit statements on research goals in the revised manuscript (lines 69 - 77).

"The primary aims of this investigation are to understand (1) the long-term annual and seasonal variation of the methane budget for a typical alpine permafrost site in the QTP, and (2) the environmental factors controlling these CH4 variations and possible underlying mechanisms. In addition, while the consumption and production of ecosystem methane are known through microbial activities, conventional investigations on seasonal methane fluxes usually used climate or vegetation defined "seasons". Therefore, a third research goal of this current study is to investigate if the classical vegetation productivity-based definition of growing season will be useful for defining the methane flux seasonality."

2. ii) Cryoturbation is a collective term for many soil transport processes based on freezing and thawing that lead to a subduction of topsoil horizons down to the transition zone. This term is wrongly used in this manuscript. I think what the authors mean is freezing and thawing instead. Please, correct this.

*Response:* Thanks for pointing out the unclear use of the term cryoturbation. We have replaced "cryoturbation" with "freezing and thawing" throughout the manuscript, following your suggestion.

3.iii) In addition to eddy covariance based observations of methane fluxes, this study presents a lot of observations of environmental factors, such as climate and soil properties. This is really interesting because this allows addressing the question on why do we see this strong seasonality with methane sources (summer, autumn) and sinks (winter, spring). However, the data presented does not explain these seasonal differences. The attempts of explanation in the discussion section with several hypotheses are important but please include into this discussion how these hypotheses could have been proven by your data or which other measurements were required.

*Response:* Thanks for your constructive comments. We completely agree with the reviewer on the importance of bringing the measured environmental data to explain results and test hypotheses in the discussions of the seasonal CH4 flux difference. In this revised work, we tried to use our field measured environmental data in the discussion section for hypothesis testing and explanations.

Following this suggestion, we have revised the related discussions by emphasizing the explanation and hypotheses of seasonal CH4 fluxes with environmental data. In particular, we also supplied a new table of seasonal soil water content variation (Supplementary Table 1) and a new figure of soil temperature data of spring\_ and autumn\_ (Supplementary Figure 10), which are important for explaining the observed seasonal variations in CH4 fluxes. Other major revisions that make connections between our field observation data and hypothesis testing are summarized below.

(1) To explain the hypothesis of different soil hydrothermal conditions will greatly influence CH4 cycles in permafrost regions, we supplemented data of the snow-cover time and SWC of 0-40cm (Supplementary Table 1) within footprint when discussed the CH4 flux during winter. The data showed that the QTP alpine steppe generally has little to no snow cover during winter. And this relative dry soil could facilitate the oxidation of CH4, and therefore reduce the size of the winter CH4 source when compared to other permafrost regions, in this revised manuscript, on lines 461-463.

- (2) Before we invoked the hypothesis of *seasonal* variations in soil thawing and freezing dynamics to explain observed spring\_ and autumn\_ CH4 flux variation, we first ruled out the possible cause from *spatial* variation in vegetation species, soil type and soil water content by providing data and a new photo of the footprint of the eddy covariance tower (Supplement Figure 11). This photo clearly shows the spatial composition of the entire footprint is relatively homogeneous; and therefore, indirectly support our hypothesis that the observed seasonal CH4 flux variations are likely caused by seasonal differences in soil thawing and freezing dynamics. On lines 501 -508 and 969 970.
- (3) We also provided some freezing-thawing process details with pointing-to-data in explaining observed CH4 patterns. For example, on lines 490 - 496.

Finally, the new discussion of seasonal variations in CH4 fluxes can be found in Lines 451 - 527 of the revised manuscript.

4. In general, the temporal differences in eddy cov CH4 time series could have been due to either temporal differences in soil processes or spatial differences of the footprint. I strongly suggest to first rule out the latter case before discussing all kind of soil processes leading to the seasonality of methane fluxes: Are there more wet or dry soil areas in the footprint and do we see methane flux dynamics due to changes in the footprint? Please, analyze wind direction and wind speed together with methane fluxe in Fig. 5. Also, the main wind direction can be displayed in color scale in fig 9. **Response:** Thank you for your constructive comments. We fully agree with the reviewer that because of the changing prevalent wind directions among different seasons, the spatial variation of the measured footprint may have affected the seasonal interpretation of methane fluxes. As our field picture (Supplementary Figure 11) showed, the footprint was generally spatially homogeneous in terms of vegetation and soil types. We have added the content of the spatially homogeneous vegetation and soils in the discussion of the seasonal methane fluxes variations (Lines 501 - 508 of the revised manuscript), and we supplemented the seasonal soil water content data in the revised manuscript (Supplementary Table 1).

---

## Editor Decision (ED1)

[revised manuscript text omitted]

160cm can up to 21.0% (Supplementary Table 1), however, soils are bidirectionally frozen from both top (ground surface) and bottom, the permafrost table, which is about 200~400 cm deep ; (Supplementary Figure 8; Wu and Zhang, 2010a), doesn't form a layer of thin ice during the nighttime surface soil freezing, because on the one hand, the frozen soil of the ground surface (about 0-40cm) prevents the outside liquid water from permeating. On the other hand, the freezing itself will reduce the liquid water content in the soil (Ma et al., 2015) Therefore, it creates finely closed anaerobic gaps that allow $CH_4$ and $O_2$ gases into deep soils (about

50~400cm; Mastepanov et al., 2008; Mastepanov et al., 2013; Zona et al., 2016). Meanwhile, the temperature of deep soils (about 50~400cm), still remains at a relatively high level (Supplementary Figure 10; b), and methanotrophic bacteria there are still active at this high Tsoil (Figure 2). This could be one important mechanism for autumn— soil $CH_4$ consumption. In addition, in principal it was also possible that the observed seasonal variation in $CH_4$ flux may actually arise from the spatial variation of the footprint covered by the eddy covariance site (within 175m), given that prevalent wind direction changes seasonally (Supplementary Figure 4).

Nonetheless, we found that the same vegetation species and soil exist in different directions to the tower within the footprint (Supplementary Figure 11). Hence this spatial vegetation and soil homogeneity further confirm that seasonal soil freezing and thawing differences may likely be the main explanation for seasonal $CH_4$ variations.

Furthermore, we suggested that the specific autumn— soil vertical structure may help explain why the site was a $CH_4$ sink, unlike the $CH_4$ source in spring—. The sequential probing data enables us to establish a rough estimate on the soil vertical structure during the autumn—

thawing–freezing process, in which the vertical distribution of clay, sandy soils, and soil organic layers was mixed like a multi–layer hamburger structure, rather than forming a gradual change (Figure 6: e). As the soil profile has a different density, thermal conductivity, heat of phase transition, salinity of soil, and so on, we boldly conjecture that, similarly, the 
[revised manuscript text omitted]

Tsoil of 100cm, 200cm, 450cm  below -2 °C and during autumn the Tsoil of 40cm almost overlap  Tsoil of 50cm to make the figure more  we removed the Tsoil of

100cm, 200cm, 450cm in figure (a) and removed the Tsoil of 40cm for figure (b).

[Figure]

**Supplementary Figure** 11. A bird's eye view of the eddy covariance in Beilu'he station

---

## Author Response (AR2)

Dear Dr. Hauck,

We thank you very much for your thorough reading of our manuscript for grammatical/language issues, which we have accepted most of your suggested changes or have rephrased according to your advices. We also carefully check the entire manuscript and correct some other minor typos and mistakes. You can find all the changes in the revised final manuscript in blue and green in the track change version. The only thing we do not change as you suggested is on the use of "diel". We intend to keep "diel" instead of replacing it with "diurnal". According to the Merriam-Webster dictionary, "diel" involves "a 24-hour period that usually includes a day and the adjoining night", but "diurnal" can be ambiguous since it sometimes can refer to daytime. In addition, the use of "diel" in geosciences is not uncommon (for example, Burns et al., The influence of warm-season precipitation on the diel cycle of the surface energy balance and carbon dioxide at a Colorado subalpine forest site. Biogeosciences, 12, 7349-7377, 2015).

We also thank Reviewer #1 for his/her positive comments and suggestions. We have revised the manuscript according the reviewer's suggestions, including the improvement of the captions for Supplementary Figure 4, the addition of seasonal footprint changes which however may not influence the sign of $CH_4$ fluxes due to the homogeneous landscape, and the harmonization of the y-axis in Figure 4. Detailed responses to each point raised by the reviewer can be found in the attached "Responses to Reviewers" document.

We hope this revision has satisfactorily addressed all your questions and concerns and have corrected all potential problems as we can before the publication. Thank you again for your time and efforts.

Best regards,

Hanbo Yun

**Responses to Reviewers**

Responses to reviewer#1

1. *The new Supplementary Figure 4 clarifies wind speed and direction over the 4 seasons. I request improving the caption of Supplementary Figure 4 and explain sub-figures. I assume these stand for the 4 seasons winter, spring, summer, autumn?*

***Response:*** Yes, you are correct that the sub-figures of Supplementary Figure 4 stand for winter, spring, summer, and autumn, respectively. In the revised manuscript, we have improved the caption by adding the sub-figure explanations.

[Figure]

**Supplementary Figure 4.** Diurnal mean of wind speed and direction between 2012 and 2016: (a) is winter, (b) is spring, (c) is summer, and (d) is autumn. All data are presented as mean values with standard deviations (mean ± standard deviation).

*2. If so, this figure can be used in concert with Supplementary Figure 9 in order to rule out any importance of footprint changes for the sign of the methane balance. If so, please include one sentence of text into the manuscript about it.*

*Response:* Thank you for your constructive comments. We fully agree with the reviewer that seasonal changes in the footprint may have the potential influencing the sign of the methane balance. Following your suggestion, we have included the explicit information regarding the footprint of methane fluxes in different seasons in the revised manuscript (lines 351 – 354).

"Across different seasons the footprint of the monitored $CH_4$ flux changed following the change of the prevalent wind direction. In winter and spring, the major footprint was from east of the EC tower; while in summer and autumn, the major footprint was from the EC tower's west and north (Supplementary Figure 4)."

We also note in the Discussion that the footprint change may not influence the sign of methane fluxes due to landscape homogeneity in different footprints.

"Nonetheless, we found that the same vegetation species and soil exist in different directions to the tower within the footprint (Supplementary Figure 11). This spatial vegetation and soil homogeneity rules out the potential influence of footprint changes on the sign of $CH_4$ balances, and further confirms that seasonal soil freezing and thawing differences may likely be the main explanation for seasonal $CH_4$ variations." (lines 493 - 497)

*3. However, the new Supplementary Figure 4 does not match the previous figure 9 about the same content - actually they show opposite wind pattern. Please, clarify!*

***Response:*** We are sorry for the confusion about the apparent difference (but they are actually the same) between the new Supplementary Figure 4 and the previous figure 9. The previous figure 9 is a wind polar figure, for which we used flow vector (wind *blowing to*) as the x-axis. In the new Supplementary Figure 4, we used the standard wind direction (wind *blowing from*). We have noted the exact meaning of wind direction in the caption of the new Supplementary Figure 4.

*4. My last minor request is to harmonize the y-axis of the new Figure 4 such that the scale is equal in all sub-plots.*

***Response:*** Thanks for your constructive comments. Follow your comment, we have harmonized the y-axis of Figure 4 (copied below).

[Figure]

[revised manuscript text omitted]
 provided an average ofan estimate on the uncertainties caused by the different U* to filter outthresholds. The second uncertainty source was due to insufficient power supply. In this research, all instrument power was supplied by solar panels. Extended periods of rainy, cloudy, and snowy weather, would cause the instrument to stop working due to an insufficient power supply. When we used the method to fill the gapgap-filling method mentioned above, it would cause the $CH_4$ flux to deviate from the true value.  To our knowledge, the $CH_4$ flux data was largely uncertain under rainy conditions.

**2.6 Based on microbial activities classification New classification system of the four seasons based on microbial activities classification**

We redefined the four seasons of spring spring, summer summer, autumn autumn, and winter winter, and based on the microbial activity parameters of the new seasons on microbial activities (Figure 2), ALT variety variability coefficients (ALT variabilityvariety coefficient = ($ALT_{i+1}$ - $ALT_i$) / $ALT_{Max}$, where $ALT_{Max}$ is the maximum of ALT per year), and $T_{soil}$. Below, we describe the start date of each season (The the end date of a season is the day immediately before the start of the next season).

Spring Spring starts at the first day of two consecutive observation periods fulfilling both (1) ($\Delta II$ + $\Delta I$) / 2 $\geq$ 15%, and (2) the ALT variabilityvariety coefficient $\geq$ 0.05.

Summer Summer starts on the first day of two consecutive observation periods when (1) ($\Delta II$ + $\Delta I$) / 2 $\geq$ 45%, (2) ALT variabilityvariety coefficient $\geq$ 0.35, and (3) five successive days with $T_{soil}$ at 40 cm soil depth $\geq$ 0 °C.

Autumn starts on the first day of two consecutive observation periods when (1)

$(\Delta II + \Delta I) / 2 \geq 55\%$, (2) the ALT variability coefficient $\geq 0.60$, and (3) five successive days the $T_{soil}$ of 10 cm $< 5\ °C$.

Winter starts on the first day of two consecutive observation periods that (1) $(\Delta II +$

$\Delta I) / 2 < 15\%$ and the ALT variability coefficient $< 0.05$.

To test the robustness of our new seasonal division method in our methane cycle analysis, we compared empirical $CH_4$ flux estimates using different season definitions (Table 2). In addition to our new method that was based on top soil microbe activity, $T_{soil}$ of 0 – 40 cm, and permafrost active layer variability (hereafter refer to as SMT), we also used three conventional methods. based on (i) vegetation cover and temperature change (VCT), (ii) based on

Julian months (JMC), and (iii) based on vegetation phenology change (VPC).

The VCT method splits a year into a plant growing season and a non–growing season; the JMC method assumes May to October as a plant growing season, and November to the following April as a non–growing season; and the VPC method defines a plant growing season as the period between the time when all dominant grass species (*Carex*

*moorcroftii* Falc. ex Boott, *Kobresia tibetica* Maxim, *Androsace tanggulashanensis*, *Rhodiola*

*tibetica*) germinate and that when they all senesce.

**2.7 Statistical Analyses**

To understand the connections between $CH_4$ fluxes and associated environmental factors, we performed a series of statistical analyses, including correlation, principal component analyses (PCA), and linear regression analyses, in IBM SPSS (IBM SPSS Statistics 24; IBM, Armonk NY,

USA). Specifically, we used bivariate correlation to examine pairwise relationships between environmental factors and $CH_4$ fluxes. We also used PCA and linear regressions to explore the sensitivity of $CH_4$ fluxes to simultaneous environmental fluctuations in wind speed, $T_{air}$, air relative humidity, Rn, vapor pressure deficit (VPD), albedo, SHF, SWC, and $T_{soil}$. Before performing PCA and linear regressions, the entire dataset was examined for outliers (Cook's Distance, < 0.002), homogeneity of variance (Levene test, $p < 0.05$), normality (Kolmogorov–Smirnov test, smooth line for histogram of Studentized residuals), collinearity (variance inflation factor, $0 < VIF < 10$), potential interactions ($t$–test, $p < 0.05$), and independence of observations ($t$–test, $p < 0.05$).

We performed structural equation modeling (SEM) to evaluate the effects of environmental variables on $CH_4$ fluxes for different seasons. SEM is a widely-used multivariate statistical tool that incorporates factor analysis, path analysis, and maximum likelihood analysis. This method uses *priori* knowledge of the relationships between focus variables to verify the validity of hypotheses. Here we performed SEM analyses with AMOS 21.0 (Amos Development Corporation, Chicago, IL, USA). All data are presented as mean values with standard deviations.

**3. Results**

**3.1 Meteorological Conditions**

We first reported the statistics of the meteorological conditions environmental factors at the Beilu'he Permafrost Weather Station based on meteorological records frombetween 2012 to 2016. Mean annual $T_{air}$ was -4.5 ℃ (Supplementary Figure 1), with minimum and maximum mean dieldiurnaldiel temperatures of -21.6 ℃ (12th January, 2012) and 13.8 ℃ (28th July, 2015), respectively. Average net radiation was 82.8 Wm$^{-2}$, while with the maximum was 
[revised manuscript text omitted]

**Comment [CA1]:** Change "Spring_, Autumn_…" in the figure legend to "Spring, Autumn, …"

**Comment [CA2]:** Also change the x-axis name to "Month" instead of "Date (Month)".

[Figure]

[Figure]

**Figure 3.** Annual patterns of diurnael methane (CH$_4$) flux and precipitation variations from 2012 to 2016. Positive values indicate CH$_4$ release and negative values indicate CH$_4$ uptake by ecosystems. Red dots and light green lines are CH$_4$–C flux variation, and the deep blue histograms show diurnael precipitation accumulation. Pink, olive, cyan, and orange blocks mean spring, summer, autumn, and winter seasons respectively, according to our new method of SMT (see Methods). Black, cyan, and pink dotted lines with bars separated the plant growing from non–growing seasons and stand for seasons by the method JMC, VCT, and VPC, respectively. Details about the methods JMC, VCT, and VPC can be found in section 3.2.

[Figure]

[Figure]

**Figure 4.** Diurnacl CH4 fluxes from 2012 to 2016 for different seasons. Blue, pink, green and orange,
represent winter, spring, summer, and autumn, respectively; (a1), (a2), (a3), and
(a4) are for 2012; (b1), (b2), (b3), and (b4) are for 2013; (c1), (c2), (c3) and (c4) are for 2014; (d1), (d2), (d3),
and (d4) are for 2015; (e1), (e2), (e3), (e4) and (f1) are for 2016.

[Figure]

**Figure 5.** Regression comparison between observation and modeled methane fluxes with four different seasonal definitions and classification models. Panels (a), (b), (c), and (d) are for the SMT, JMC, VCT, and VPC

methods, respectively.

                                                          -

[Figure]

**Figure 6.** Location of exploratory pits and drillings in this study in autumn: (f) is photo of a typical ground surface (October 16th, 2014). Green flags represent the location for the soil survey by test pitsting and drilling.

(a), (b), (c), and (d) show soil profiles of 0 − 250 cm depths  at the North (1), South (2), East (3), and West (4) corners of the eddy covariance footprint , respectively. (a1), (b1), (c1), and (d1) are drilling cores, with clear ice (white) in (a1), (b1), and (d1), but not in (c1); (e) provides an illustration that combines results from drillings, test  pits and multi–channel ground–penetrating radar (Malå Geoscience, Sweden) for active layer variations in permafrost area during the autumn season; and (e1) is a core sample of the same drilling (October 16$^{th}$, 2014).

**Supplement**

**Supplementary Table 1** Seasonal soil water content (SWC, %) of  winter,  spring,  summer, and  autumn from 2012 to 2016.

| Seasonal | Period | 10 cm | 20 cm | 40 cm | 80cm | 160cm |
|---|---|---|---|---|---|---|
| | | Soil Water Content (SWC), % | | | | |
|  Winter | 2012 early | 0.11 | 0.08 | 0.07 | 0.11 | 0.14 |
| | 2012-2013 | 0.10 | 0.08 | 0.07 | 0.11 | 0.16 |
| | 2013-2014 | 0.10 | 0.08 | 0.07 | 0.11 | 0.13 |
| | 2014-2015 | 0.10 | 0.08 | 0.07 | 0.11 | 0.17 |
| | 2015-2016 | 0.10 | 0.08 | 0.07 | 0.11 | 0.16 |
| | 2016 later | 0.10 | 0.08 | 0.07 | 0.12 | 0.19 |
| | Average | 0.10 | 0.08 | 0.07 | 0.11 | 0.16 |
|  Spring | 2012 | 0.13 | 0.09 | 0.08 | 0.11 | 0.13 |
| | 2013 | 0.12 | 0.09 | 0.08 | 0.11 | 0.13 |
| | 2014 | 0.12 | 0.08 | 0.07 | 0.11 | 0.13 |
| | 2015 | 0.13 | 0.09 | 0.08 | 0.11 | 0.14 |
| | 2016 | 0.12 | 0.09 | 0.08 | 0.13 | 0.15 |
| | Average | 0.12 | 0.08 | 0.08 | 0.11 | 0.14 |
|  Summer | 2012 | 0.18 | 0.11 | 0.10 | 0.17 | 0.27 |

| | | | | | | |
|---|---|---|---|---|---|---|
| | 2013 | 0.16 | 0.11 | 0.11 | 0.19 | 0.25 |
| | 2014 | 0.16 | 0.10 | 0.10 | 0.16 | 0.24 |
| | 2015 | 0.16 | 0.10 | 0.10 | 0.19 | 0.28 |
| | 2016 | 0.16 | 0.10 | 0.09 | 0.18 | 0.28 |
| | Average | 0.17 | 0.10 | 0.10 | 0.18 | 0.26 |
|  Autumn | 2012 | 0.14 | 0.09 | 0.08 | 0.14 | 0.21 |
| | 2013 | 0.14 | 0.09 | 0.09 | 0.15 | 0.20 |
| | 2014 | 0.16 | 0.10 | 0.10 | 0.16 | 0.22 |
| | 2015 | 0.15 | 0.10 | 0.09 | 0.15 | 0.21 |
| | 2016 | 0.16 | 0.10 | 0.09 | 0.16 | 0.21 |
| | Average | 0.15 | 0.10 | 0.09 | 0.15 | 0.21 |

[Figure]

**Supplementary Figure 1.** Air temperature ($T_{air}$) measured  3 meters above the ground surface:

(a), (b), (c), and (d) are half–hour  mean values in spring, summer, autumn, and winter, respectively; (e) shows diurna–scale mean values from to 2016.

[Figure]

**Supplementary Figure 2.** Net radiation (Rn)  measured 3 meters above the ground surface:

(a), (b), (c), and (d) are half–hour  mean values in spring, summer, autumn, and winter, respectively; (e) shows diurnal–scale mean values from to 2016.

[Figure]

**Supplementary Figure 3.** Vapor pressure deficit (VPD)  measured 3 meters above the ground surface: (a), (b), (c), and (d) are half–hour  mean values in spring, summer, autumn, and winter, respectively; (e) shows diurna mean values from

2012 to 2016.

[Figure]

[Figure]

**Supplementary Figure 4.** Diurnael mean of wind speed and direction between 2012 and

2016: (a) is winter, (b) is spring, (c) is summer, and (d) is autumn.  Note the direction of wind means the direction wind blows *from.* All data are presented as mean values with standard deviations (mean ± standard deviation).

[Figure]

**Supplementary Figure 5.** Comparison between soil water content (SWC) of two different time resolutions from 2012 to 2016, (a) is the half–hourly scale SWC at soil depths of 10 cm, 20 cm,

40 cm, 80 cm, and 160 cm; and (b) is the 4–hourly mean SWC for the same depths.

[Figure]

**Supplementary Figure 6.** Half–hour measurements scale of 0 − 200 cm soil temperature (T$_{soil}$)

variations from 2012 to 2016, (a) is for soil depths of 0 cm, 5 cm, 10 cm, 20 cm, 30 cm, 40 cm,

50 cm, (b) is for soil depth of 70 cm, 80 cm, 100 cm, 150 cm, 160 cm, and 200 cm.

[Figure]

**Supplementary Figure 7.** Soil heat flux (SHF) at depth of 5 cm and 15 cm: (a), (b), (c), and (d)

are half–hour scale-mean values in spring, summer, autumn, and winter, respectively; (e) shows diurnae scale mean values from 2012 to 2016.

.

[Figure]

**Supplementary Figure 8.** Characteristics of the seasonal freezing and thawing processes of the active layer for years: 2012, 2013, 2014, 2015, and 2016. Different colors represent the soil temperature gradients from -16 °C to 20 °C. The depth of 0 °C⎵ represent the active layer thickness (ALT).

[Figure]

**Supplementary Figure 9.** Seasonal $CH_4$ rate mean value from 2012 to 2016: (a) is winter, (b) is spring, (c) is summer, and (d) is autumn. In the (a), 2012E is started from January 1st, 2012 and ended on February 17th, 2012; 2012W is started from 19th November, to 4th February, 2013; 2013W is started from 1st December, 2013 to 17th February, 2014;

2014W is started from 6th November, 2014 to 4th February, 2015; 2015W is started from 9th

November, 2015 to 15th February, 2016; 2016L is started from October 26th, 2016 and ended on

December 31st, 2016. All data are presented as mean values with standard deviations (mean ±

standard deviation).

[Figure]

**Supplementary Figure 10.**  Mean half–hour values  of 0 − 450 cm soil temperature ($T_{soil}$)  from 2012 to 2016, (a) is for  spring, (b) is for

 autumn. Note, that during spring $T_{soil}$ of 100cm, 200cm, 450cm is always

 below -2 °C and dring  autumn the $T_{soil}$ of 40cm almost overlap to $T_{soil}$  with 50cm

  To make the figure more readable,  we removed the $T_{soil}$ values of 100cm, 200cm,

450cm in figure (a) and removed the $T_{soil}$ values of 40cm for figure (b).

[Figure]

**Supplementary Figure** 11. A bird's eye view of the eddy covariance site in at the Beilu'he

                                    station